

# Methane (CH$_4$) and nitrous oxide (N$_2$O) from ground-based FTIR at Addis Ababa: observations, error analysis and comparison with satellite data.

Temesgen Yirdaw berhe[1], Gizaw Mengistu Tsidu[1,2], Thomas Blumenstock[3], Frank Hase[3], and Gabriele P. Stiller[3]

[1] Department of Physics, Addis Ababa University, P.O.Box 1176, Addis Ababa, Ethiopia

[2] Department of Earth and Environmental Sciences, College of Science, Botswana International University of Technology and Science (BIUST), Priv.Bag 16, Palapye, Botswana

[3]Institute of Meteorology and Climate Research (IMK-ASF), Karlsruhe Institute of Technology (KIT), Karlsruhe, Germany.

**Correspondence:** T. Yirdaw Berhe (temiephys@gmail.com)

**Abstract.** A ground based high spectral resolution Fourier transform infrared (FTIR) spectrometer has been operational at Addis Ababa (9.01° N, 38.76° E, 2443 m a.s.l) since May 2009 to obtain information on the total column abundances and vertical distribution of various constituents in the atmosphere. The retrieval strategy and the results on information content and corresponding full error budget evaluation for methane and nitrous oxide retrievals are presented. They reveal the high quality of

FTIR measurements at Addis Ababa. The FTIR products of CH$_4$ and N$_2$O have been compared to coincident volume mixing ratio (VMR) measurements obtained from the reduced spectral resolution (Institute of Meteorology and Climate Research) IMK/IAA MIPAS satellite instrument (Version V5R_CH4_224 and V5R_N2O_224), the Microwave Limb Sounder on board of the Aura satellite (Aura/MLS) (MLS v3.3 of N$_2$O and CH$_4$ derived from MLS v3.3 products of CO, N$_2$O and H$_2$O) and the Atmospheric Infrared Sounder (AIRS). From comparison of FTIR CH$_4$ and IMK/IAA MIPAS V5R_CH4_224, a statistically

significant bias between -4.8 and +4.6% in altitude ranges of the upper troposphere and lower stratosphere (15-27 km) are determined. The largest negative bias in FTIR CH$_4$ is found in the altitude range of 11-19 km with a maximum difference of -0.08 ppmv (-4.8%) at around 15 km, a positive bias of less than 0.14 ppmv (9%) is found in the altitude range of 21 to 27 km with a maximum value at around 27 km with respect to AIRS. On the other hand, comparison of CH$_4$ from ground based FTIR and MLS-derived CH$_4$ (version 3.3) indicate existence of a significant positive bias of 2.3% to 11% in the altitude

range of 20 to 27 km and a negative bias -1.7% at 17 km. In the case of N$_2$O derived from FTIR and MIPAS V5R_N2O_224 comparison, a significant positive bias of less than 15% in the altitude range 22-27 km with a maximum value at around 25 km and a negative bias of -7% have been found at 17 km. A positive bias of less than 18.6% in FTIR N$_2$O for the altitude below 27 km is noted when compared to MLS v3.3 N$_2$O. Precision of ground based FTIR CH$_4$ and N$_2$O in the upper troposphere and lower stratosphere over Addis Ababa are better than 7.2% and 9%, respectively which are comparable to the bias obtained

from the comparisons.



# 1    Introduction

Methane ($CH_4$), nitrous oxide ($N_2O$) and chlorofluorcarbons (CFCs) are tropospheric species which are the main source gases to the chemical families $NO_x$, $ClO_x$, and $HO_x$ (Jacobson, 2005). The reaction of $CH_4$ with hydroxyl radicals reduces ozone in the troposphere and it influences the lifetime or production of other atmospheric constituents such as stratospheric water

vapour and $CO_2$ (Michelsen et al., 2000; Boucher et al., 2009), whereas the lifetime of $N_2O$ is determined by its rate of UV photolysis or reaction with $O(^1D)$ (Collins et al., 2010).

Methane retrievals from near-infrared spectra recorded by the SCIAMACHY instrument on board ENVISAT suggested unexpectedly large tropical $CH_4$ emissions and the impact of water spectroscopy on methane retrievals with the largest impacts in the tropics (Frankenberg et al., 2008). The recent increasing impact of $CH_4$ and $N_2O$ to the global warming has also been

assessed by the last AR4 IPCC report (IPCC, 2007; Sussmann et al., 2012). Nitrous oxide ($N_2O$) becomes the dominant ozone depleting substance emitted in the $21^{st}$ century (Ravishankara et al., 2009). In 2007 and 2008, IASI observed an increase of mid-tropospheric methane in the tropical region of $9.5 \pm 2.8$ and $6.3 \pm 1.7$ ppbv $yr^{-1}$ respectively (Crevoisier et al., 2012). Long lived compounds ascend in the tropics, cross the tropical tropopause and are subsequently redistributed by the Brewer-Dobson circulation (Holton, 2004). According to the World Meteorological Organization (WMO), 2010 report (WMO, 2010), 96% of

the increase in radiative forcing is due to the five long lived green house gases: carbon dioxide, methane, nitrous oxide, CFC-12 and CFC-11. The sources and sinks of atmospheric methane ($CH_4$) and its budget in the tropics are not yet well quantified and have large uncertainty. Which is due to the scarcity of measurements (e.g. Meirink et al. (2008b)).

Tropics is the location where two important exchange processes in the atmosphere are taking place, the inter hemispheric exchange and the entry of tropospheric air mass into the stratosphere (Petersen et al., 2010; Fueglistaler et al., 2009). Composi-

tion of tropical atmosphere also plays a critical role for stratospheric chemistry (Solomon, 1999; IPCC, 2007). Measurements and interpretation of atmospheric trace gas composition of tropics is vital for a better understanding of the budgets, sources and sinks of trace gases in the atmosphere and their effects on atmospheric chemistry, greenhouse effect and climate changes globally. Emissions within the tropics contribute substantially to the global budgets of many important trace gases (IPCC, 2007; Frankenberg et al., 2008).

The ground-based FTIR measurement at the Addis Ababa site has been launched since 2009 in collaboration with Karlsruhe Institute of Technology, Germany to measure concentrations of various trace gases in the lower and middle atmosphere over Addis Ababa. The quality of ground based FTIR measurements of atmospheric trace gases and their use to understand various lower and middle atmospheric processes have been reported in a number of previous studies (Kenea et al., 2013; Mengistu Tsidu et al., 2015; Schneider et al., 2015, 2016; Barthlott et al., 2017). $H_2O$ VMR profiles and integrated column amounts

from ground based FTIR measurements of the Addis Ababa site were also compared with the coincident satellite observations of Tropospheric Emission Spectrometer (TES), Atmospheric Infrared Sounding (AIRS) and Modular Earth Sub model System (MESSy) model and the result confirmed reasonably good agreement (Kenea, 2014). Laeng et al. (2015) found the MIPAS $CH_4$ profiles V5R_CH4_222 below 20 to 25 km biased high and provided +14% as the most likely bias. For a later and improved



data version, namely V5R_CH4_224/225, Plieninger et al. (2016) found a positive bias between 0.1 and 0.2 ppmv. For the MIPAS $N_2O$ data version V5R_N2O_224/225, Plieninger et al. (2016) determined the bias to be between 0 and +30 ppbv.

In this study, the previous work on intercomparison is extended to source gases $CH_4$ and $N_2O$ from ground-based FTIR. Intercomparisons of vertical profiles and column amounts retrieved from solar spectra observed by the Fourier Transform Spec-
trometer at the Addis Ababa site with satellite observations (MIPAS, MLS and AIRS) were made. The observed differences between ground-based FTIR and satellite observation of $CH_4$ and $N_2O$ are analysed using the statistical tools detailed in von Clarmann (2006). The measurement site and the FTIR spectrometer along with the retrieval approach will be introduced in Section 2 and the retrieved information content and spectral analysis will be discussed in Section 3. A short description of satellite measurement techniques followed by the detailed intercomparison with satellite products will be presented in Section
4 and 5 respectively. Finally, a summary and conclusions are given in Section 6.

## 2 Measurement site and Instrumentation

### 2.1 Measurement site

The Addis Ababa FTIR spectroscopy was established to acquire high-quality long-term measurements of trace gases for the purpose of understanding chemical and dynamical processes in the atmosphere and to validate models and satellite measure-
ments of atmospheric constituents. The geographic position of the observatory is 9.01° N, 38.76° E and its suitability has been confirmed from the measurements of tropical stratospheric ozone, precipitable water vapour and isotopic composition of water vapour (Kenea et al., 2013; Mengistu Tsidu et al., 2015; Schneider et al., 2015, 2016; Barthlott et al., 2017). Addis Ababa is a tropical high altitude observing site and as such extremely important to understand processes near the tropical tropopause. Physical process in tropics, mainly around tropopause layer has a vital role in climate change and the general circulation of the
tropical troposphere, which would control the transport of energy, water vapour and trace gases in the climate system derived by the deep convection (Holton and Gettelman, 2001).

### 2.2 The FTIR Spectrometer and Retrieval

Fourier transform spectroscopy has been applied very successfully to the study of trace gases in the atmosphere by examining terrestrial atmospheric absorption lines in the infrared spectrum from the Sun. Measurement of Sun's spectra at the Earth
surface provides an information about atmospheric composition. The high-resolution FTIR Spectrometer, Bruker IFS120M upgraded with 125M electronics, from the Bruker Optics Company in Germany was installed in May, 2009 at the Addis Ababa site. This technique uses the Sun as a light source to quantify molecular absorptions in the atmosphere and then retrieve trace gases abundance. The spectral coverage of the IFS120M instrument at Addis Ababa site is 750 - 4000 $cm^{-1}$ using seven filters. This interferometer is equipped with indium-antimonide (InSb) detector, which allow the coverage of the 1500 - 4400 $cm^{-1}$
spectral interval. In this spectral interval, a very large number of species that reside in the atmosphere can be detected. For the work presented in this paper, we used PROFFIT Ver 95 algorithm (Hase et al., 2004). It has been developed based on semi-





empirical implementation of the Optimal estimation Method (Rodgers, 2000) to derive the VMR profiles and column amounts of $CH_4$ and $N_2O$ from measured spectra in the microwindows that span spectral range of 2400 - 3100 cm$^{-1}$ (3.3 - 4.1 $\mu$m).

This algorithm simulates the spectra and the Jacobians by the line-by-line radiative transfer model PRFFWD (PRoFit For-WarD model) (Hase et al., 2004). The vertical profiles over Addis Ababa have been obtained by fitting five and four selected

spectral regions (microwindows) for $CH_4$ and $N_2O$ respectively. The retrieved state vector contains the retrieval volume mixing ratios of the target gas defined in 41 layers of the tropical atmosphere. The retrieved profiles of $CH_4$ and $N_2O$ were derived using a Tikhonov-Phillips regularization method and performed on a logarithmic scale

The Optimal Estimation Method allows to characterise the retrievals, i.e., the vertical resolution of the retrieval, its sensitivity to the a priori information and degree of freedoms for signal (DOFs) quantitatively (see details in Rodgers (2000)). The

retrieved state vector $\hat{x}$ is related to the a priori ($x_a$) and the true state vectors ($x$) by the following mathematical expression

$$\hat{x} = x_a + \hat{A}(x - x_a) + \varepsilon \qquad (1)$$

where $\hat{A}$ is averaging kernel matrix and $\varepsilon$ is the measurement error. Moreover, actual averaging kernels matrix depends on several parameters including the solar zenith angle, the spectral resolution and signal to noise ratio, the choice of retrieval spectral microwindows, and the a priori covariance matrix $S_a$. The elements of averaging kernel for a given altitude gives

the sensitivity of retrieved profiles at which the real profile is present and its full width at half maximum is a measure of the vertical resolution of the retrieval at that altitude (Rodgers and Connor, 1990). Error estimation analysis is based on the analytical method suggested by Rodgers (2000):

$$\hat{x} - x = (A - I)(x - x_x) + GK_b(b - b_a) + G\varepsilon \qquad (2)$$

The averaging kernel matrix can be defined as A = GK, $I$ is the identity matrix and $G$ is gain matrix that represents the sensitivity

of retrieved parameters to the measurement, $K_b$ the sensitivity matrix of the spectrum to the forward model parameters $b$. Since we do not know the true state of the atmosphere, we cannot specify the actual retrieval error but we can only make a statistical estimate of it, which is expressed in terms of a covariance matrix. The total error in the retrieved profile can be described as a combination of measurement error and forward model parameter error. It has been suggested by Rodgers (2000) to include smoothing error to the total error budget but this concept has been revised by von Clarmann (2014)

## 3 Information content and error analysis

### 3.1 Spectroscopic data and a priori profiles

In our retrieval set-up, the profiles of $CH_4$ and $N_2O$ were retrieved, while the profiles of interfering species (see Table 1) were scaled. The a priori profiles are based on data sets from the Whole Atmosphere Community Climate Model (WACCM, http://www2.cesm.ucar.edu/working-groups) as recommended by the NDACC/IRWG (Infrared Working Group). Daily Pro-

files of pressure and temperature were taken from the NCEP reanalysis available through the NASA Goddard Space Flight Centre auto mailer from $http : //hyperion : gsfc : nasa : gov/Dataservices/automailer/index : html$. The spectroscopic



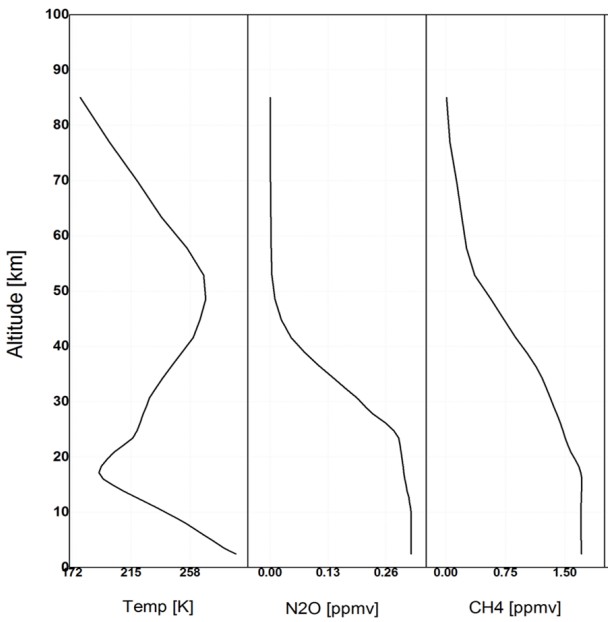

**Figure 1.** Tropics a priori profiles of $N_2O$ and $CH_4$ (second from left to right) and their expectation variation along the height of temperature (left).

parameters were taken from the High Resolution Transmission (HITRAN) database version 2004 with the 2009 and 2012 updates (Rothmann et al., 2004, 2013). Fig. 1 shows a priori profiles of $N_2O$ and $CH_4$ for tropical atmospheric conditions along with a temperature profile. Both methane ($CH_4$) and nitrous oxide ($N_2O$) are well-mixed in the troposphere and their VMR decrease with height and becomes negligible with no variation above 55 km. The vertical variability of $N_2O$ and $CH_4$ in the

stratosphere is characterized by a large vertical gradient.

The spectral micro-windows used for the retrieval are selected such that the absorption features of the target species along with a minimal number of interfering absorption lines are presented. The microwindows have been adopted from different sources. The microwindows as well as interfering gases for the two target species in this paper are shown in Table 1. However, the microwindows are somehow modified for tropics from the wondows recommended by NDACC (Network for the Detection

of Atmospheric Composition Change) as mentioned in (www.ndacc.org).

The spectral fit and residual between measured and simulated spectra at five and four microwindows for $CH_4$ and $N_2O$ respectively are depicted in Fig. 2 and Fig. 3 for example spectra recorded on Feb 26, 2013 and Dec 31, 2009 at Addis Ababa respectively. The last column of Table 1 provides typical values for the degrees of freedom for signal (DOFS) and it indicates the possible independent pieces of information of the target gases distribution. The magnitude of residuals of spectral fits are

less than 1% with both positive and negative signs ($CH_4$: 0.4%; $N_2O$: 0.34%). An optimized retrieval strategy for tropics has been established within the framework of this paper for the retrieval of $CH_4$ and $N_2O$ by applying it first to single spectra as





**Table 1.** Microwindows, interfering gases and their DOFS listed in the table are used for the retrieval of VMR profiles and column amounts of $CH_4$ and $N_2O$ from FTIR spectra recorded at Addis Ababa.

| T.Gases | MW($cm^{-1}$) | int. gases | DOFS |
|---|---|---|---|
| $CH_4$ | (2599.8,2600.5) | | |
| | (2614.87,2615.4) | | |
| | (2650.8,2651.29) | $H_2O, CO_2, NO_2$ | 2.045 |
| | (2760.6,2761.23) | | ±0.18 |
| | (2778.22,2778.55) | | |
| $N_2O$ | (2464.2,2465.57) | | |
| | (2486.55,2488.18) | $H_2O, CO_2, CH_4$ | 3.38 |
| | (2491.86,2492.9) | | ±0.15 |
| | (2522.95,2524.1) | | |

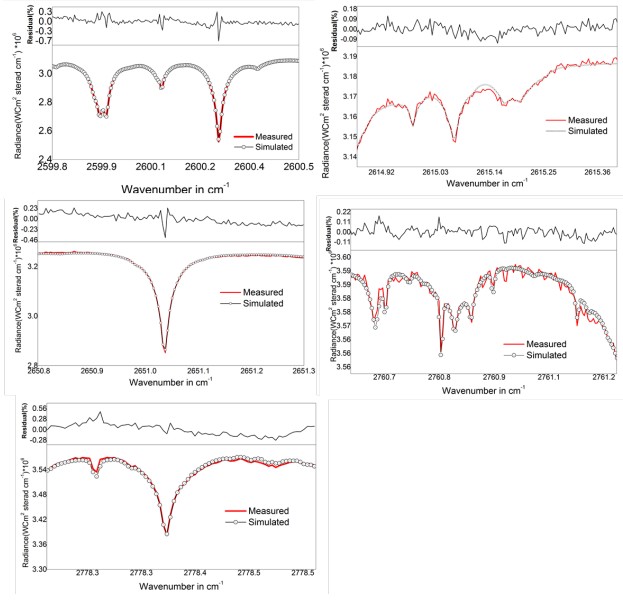

**Figure 2.** The five spectral micro-windows used for retrieval of $CH_4$, with the measured spectrum in red, the simulated spectrum in black with circle, and residuals on top of the respective microwindow. The spectrum was recorded on Feb 26, 2013, time: 101715, SD=0.1189, SEA= 69.37° , OPD=116.1, DOF = 2.23, FOV=2.27 mrad.

test cases, and later routinly to the full set of measurments. Concentrations of $CH_4$ and $N_2O$ were derived from 166 spectra of NDACC filter 3 recorded from Dec. 2009 to March, 2013.





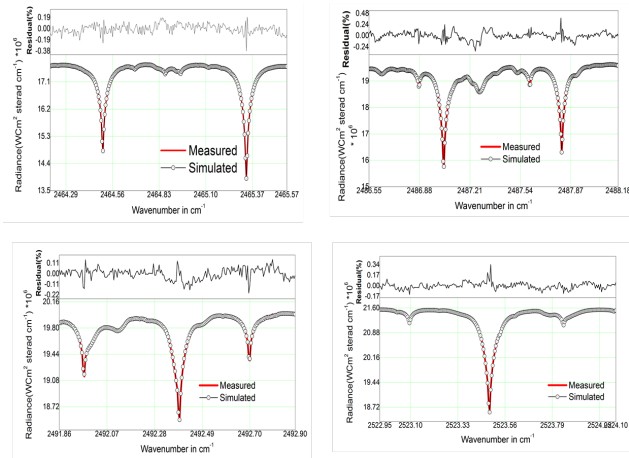

**Figure 3.** The four spectral micro-windows used for retrieval of $N_2O$, with the measured spectrum in red, the simulated spectrum in black with circle, and residuals on top of the respective microwindow. The spectrum was recorded on Dec 31, 2009, time: 093727SC, SEA= $76.58°$, OPD=100, DOF = 3.35.

## 3.2 Vertical resolution and sensitivity assessment

The averaging kernel is the most important diagnostic tool to characterize to which degree the result represents measurement or a priori information by taking the summation of individual elements of the rows of averaging kernels. Thus, $\hat{\mathbf{x}}$, which is the solution of retrieval as mathematically expressed in Eq.(1) is a combination of a priori profile $\mathbf{x}_a$ and the differences of true

values and a priori weighted by the averaging kernel matrix.

Ideally the vertical resolution of the retrieval matches with the layer spacing used for the representation of state vector. In this case the average kernel would be the identity matrix. In reality, the diagonal values of the averaging kernel matrix are below unity, indicating that at a certain altitude the retrieved value represents either a priori information or that the value of atmospheric state is influenced by a state at neighbouring altitudes. The vertical resolution is defined as full width at half

maximum (FWHM) of the rows of the averaging kernels.

Fig. 4 shows averaging kernel matrices for the retrieval of the vertical profiles of $CH_4$ and $N_2O$ mixing ratios, respectively, from the FTIR measurements at the Addis Ababa site. Fig. 4 (right panel) represents the rows of the averaging kernel matrices at selected altitudes which indicate the sensitivity of retrieved $CH_4$ and $N_2O$ values at the level to true mixing ratios. The dotted line represents the sum of all the rows of the averaging kernel, which represents the overall sensitivity of the FTIR

measurement to observe $CH_4$ and $N_2O$.

Fig. 4 (top panel) shows that the retrieval of $CH_4$ is only sensitive to the altitude range of troposphere and lower stratosphere, i.e. 2.45 km up to 27 km, since the sum of rows of **A** for all the retrieval values of $CH_4$ is greater than 0.5 up to 27 km. The trace of this averaging kernel, which is DOFS, is 2.25 for the spectra recorded on Feb. 26, 2013 and the average DOFS of the





whole data set is 2.11 ± 0.06. From this DOFS value, we can conclude that partial columns representing two different altitude ranges in the atmosphere can be obtained from the observations of $CH_4$ in tropical atmospheric conditions.

Fig. 4 (bottom panel) shows that the ground based FTIR measurement of $N_2O$ at Addis Ababa has a sensitivity larger than 0.5 from the ground to about 27 km. The amplitude of the averaging kernels indicates the sensitivity of the retrieval and the full

5  widths at half maximum (FWHM) indicate the vertical resolution of the corresponding layer. We also ignore the altitude range were the resolution of the instrument becomes beyond 20 km, which has been computed using the reciprocal of the diagonal values of averaging kernels and multiplying by the intervals of the layers as reported in Rinsland et al. (2005). The vertical resolution is less than 20 km for the altitude below around 27 km (not shown).

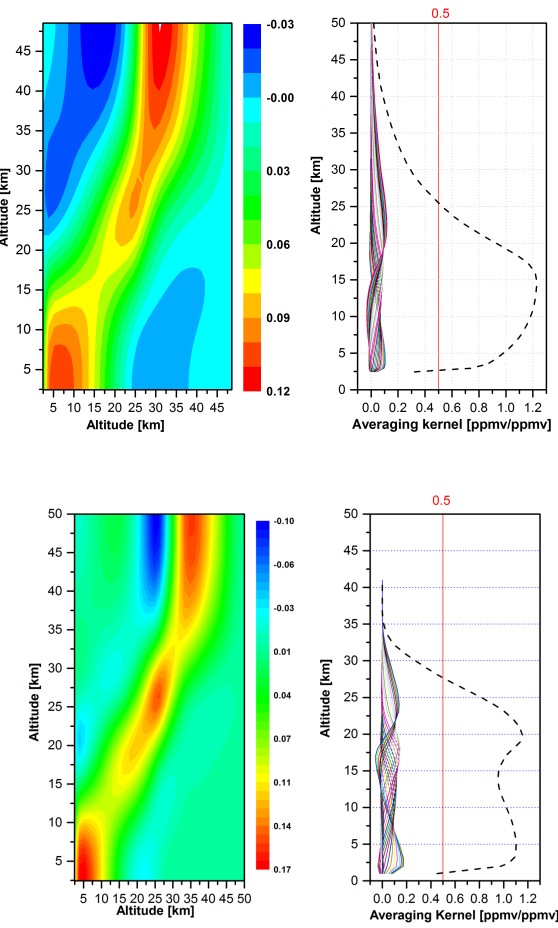

**Figure 4.** Sensitivity analysis of the retrieved profiles of $CH_4$ (top panels) and $N_2O$ (bottom panels) at Addis Ababa. The maps of the averaging kernels matrix, with their values color-coded, on the left, and selected rows of the averaging kernels as function of altitude on the right. The dotted line in the right panels is the sum of the rows of the averaging kernels.





### 3.3 Error Estimation

The error calculations conducted here are based on the error estimation package incorporated in the PROFFIT retrieval algorithm that was developed based on the analytical method suggested by Rodgers (2000). The quantified sources of errors are temperature, measurement noise, instrumental line shape, solar lines, line of sight, zero baselines offset, and spectroscopy. It

has been observed that baseline and atmospheric temperature uncertainties are the leading contribution to the total uncertainty. Details about the evaluation of the individual contributions to the error budget are provided in Senten et al. (2008). Evaluation of the individual contributions to the error budget of $CH_4$ and $N_2O$ VMRs derived from FTIR measurements at the Addis Ababa site is discussed below.

Fig. 5 shows the statistical error, systematic error and retrieved profiles (left to right) for a typical $CH_4$ (top) and $N_2O$

(bottom) retrieval from a spectrum recorded on Feb. 26, 2013 and Dec. 31, 2009 respectively. It can be noted from Fig. 5 that the main systematic error source is the uncertainty of spectroscopic parameters, whereas the major statistical error source is the baseline. Random errors are dominated by the baseline offset uncertainty and the measurement noise in the troposphere. Total estimated random error due to parameter uncertainties is depicted as dark yellow line (see Fig. 5, top panel). The total statistical error of $CH_4$ retrieval is about 0.07 ppmv (4.4%) in the lower troposphere and about 0.04 ppmv (2.25%) in the UT/LS region.

Concerning systematic errors, spectroscopic parameters are the dominant uncertainty sources and estimated total systematic error is about 0.05 ppmv (3.5%) and 0.1 ppmv (7.2%) for the lower troposphere and the UT/LS region, respectively.

Fig. 5 ( bottom panel) shows the estimated random and systematic errors for the profile retrieved from FTIR station at Addis Ababa. Random errors are dominated by the baseline offset uncertainty and temperature in the troposphere. The total statistical errors in middle and upper troposphere are between 0.009 ppmv (3.5%) and 0.03 ppmv (9%) with its major contribution from

the baseline. Spectroscopic parameters and baselines are the dominant uncertainty sources for systematic errors. The estimated total systematic error is less than 0.025 ppmv (8%) in the altitude below 22 km.

Fig. 6 shows the time series of the retrieved total column amounts (in molecules $cm^{-2}$) of $CH_4$ and $N_2O$ obtained from the Addis Ababa FTIR measurement site from 2009 - 2013. The mean total column amounts of $CH_4$ and $N_2O$ measured at Addis Ababa are $2.9 \times 10^{19}$ molecules $cm^{-2} \pm 3.4\%$ and $5.23 \times 10^{18}$ molecules $cm^{-2} \pm 6.93\%$ respectively. The sensitivity of

the observation in measuring $CH_4$ and $N_2O$ trace gases is limited to an altitude of around 27 km as explained using averaging kernel row of the measurement. The mean partial column of $CH_4$ and $N_2O$ within the sensitivity range of the instrument, which is from the surface to around 27 km, is determined as $2.85 \times 10^{19}$ molecules $cm^{-2} \pm 5.3\%$ and $5.16 \times 10^{18}$ molecules $cm^{-2} \pm 6.95\%$ respectively. The sensitivity from the averaging kernel analysis is used to determine the upper altitude limit up to which $CH_4$ and $N_2O$ data from ground based FTIR can reasonably be used. The DOFS within these partial columns limits

are about 1.03 for $CH_4$ and 1.27 for $N_2O$. Error analysis indicates that the statistical error accounts for 2.3% in the total column amounts of $CH_4$ and 2.0% in total columns of $N_2O$. Similarly, the systematic error accounts for 2.1% in total column of $CH_4$ and 2.26% in the total columns of $N_2O$. Generally, the overall contribution of both statistical and systematic errors that is total error during the retrieval of $CH_4$ and $N_2O$ from ground based FTIR are 3.1% and 3% respectively.



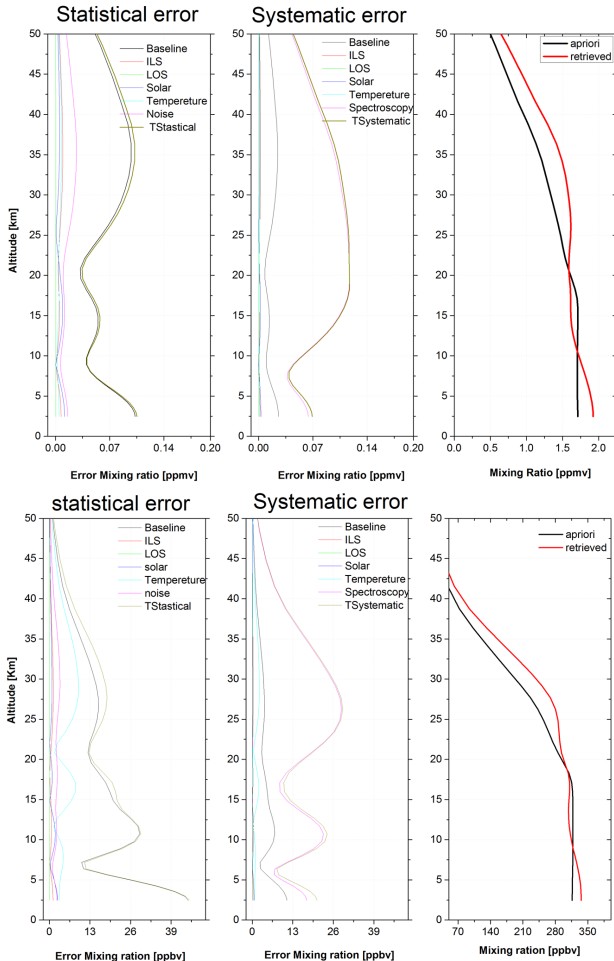

**Figure 5.** Error budgets assessment for tropical atmospheric conditions of $CH_4$ and $N_2O$ Left: Estimated uncertainty profiles for statistical error. middle: Systematic error contributions, Right: retrieved profile

## 4   Satellite measurements

### 4.1   Michelson Interferometer for Passive Atmospheric Sounding (MIPAS)

Michelson Interferometer for Passive Atmospheric Sounding (MIPAS) is a Fourier transform spectrometer for the detection of limb emission spectra from the upper atmosphere to the lower thermosphere and designed for global vertical profile mea-

5    surement of many atmospheric trace constituents relevant to the atmospheric chemistry, dynamics, and radiation budget of the middle atmosphere. Vertical resolution of MIPAS is 3-5 km depending on altitude. In this study, we have used the reduced spectral resolution (Institute of Meteorology and Climate Research) IMK/IAA MIPAS methane and nitrous oxide data product V5R_CH4_224 and V5R_N2O_224 (Plieninger et al., 2016, 2015). The analysis of the comparison between volume mixing

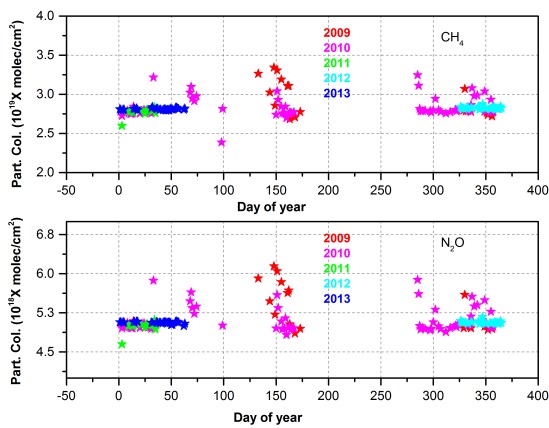

**Figure 6.** Partial columns of $CH_4$ (top) and $N_2O$ (bottom) gases over Addis Ababa in the altitude range of 2.45 to 27 km.

ratio values derived from FTIR and MIPAS were performed for the data sets between March 2009 to December 2010. MIPAS profiles points, where the diagonal element of the averaging kernels above 0.03 and the visibility flag is 1 have been used (Plieninger et al., 2016). In the stratosphere, resolution of the data products ranges from 2.5 to 7 km (Plieninger et al., 2015).

A comparison of MIPAS IMK/IAA product versions V5R_CH4_224 and V5R_N2O_224 with profiles measured by other in-
5  struments can be found in Plieninger et al. 2016. Laeng et al. 2015 had reported that MIPAS V5R_CH4_222 profiles are biased high (14%) below 20-25 km. The retrieval setup for the new MIPAS-ENVISAT $CH_4$ and $N_2O$ profiles versions V5R_CH4_224, V5R_CH4_225, V5R_N2O_224 and V5R_N2O_225 have been improved leading to reduced positive bias below 25 km with respect to other instruments (Plieninger et al., 2015, 2016).

### 4.2 Microwave Limb Sounder (MLS)

10  The Earth Observing System (EOS) Microwave Limb Sounder (MLS) is one of four instruments on the NASA's EOS Aura satellite, launched on July 15, 2004 into a near polar sun-synchronous orbit at 705 km altitude (Schoeberl et al., 2006). The MLS measures $N_2O$ in spectral region, 640 GHz from the stratosphere into upper troposphere (Waters, 2006). The spatial coverage of this instrument is nearly global (-82° S to 82° N) and individual profile spaced horizontally by 1.5° or 165 km along the orbit track. Roughly the satellite covers this latitudinal bands with 15 orbits per day or around 3500 vertical profiles
15  per day. The vertical resolution is between 4 to 6 km for $N_2O$. This instrument ascends equatorial region at local time of around 13:45 hour.

In this work, we have used version 3.3 MLS of $N_2O$ data set to validate ground-based FTIR results. However, the methane ($CH_4$) data contain vertical profiles between 100 and 0.1 hPa pressure which are derived using coincident measurements of atmospheric water vapor ($H_2O$), carbon monoxide (CO) and nitrous oxide ($N_2O$) from the EOS MLS (Earth Observing System





Microwave Limb Sounder) instrument on the NASA Aura satellite and detail are given in Minschwaner et al. (2015). Selection criteria were implemented as stated in Livesey et al. (2013). More details regarding the MLS experiment and data screening are provided in the above references in detail and at $http://mls.jpl.nasa.gov/data/datadocs.php$. Nitrous oxide derived from MLS v2.2 has been validated in Lambert et al. (2007). The authors reported that MLS $N_2O$ precision is 24-14 ppbv (9-41%)

and the accuracy is 70-3 ppbv (9-25%) in the pressure range 100-4.6 hPa.

## 4.3    Atmospheric Infrared Sounder (AIRS)

Operating in nadir sounding geometry, the Atmospheric Infrared Sounder (AIRS) on board the Aqua satellite launched into Earth orbit in May 2002 and it provides information on the vertical profiles of atmospheric temperature and water vapour from the surface to the upper troposphere i.e., up to altitudes corresponding to the 150 hPa pressure level. AIRS is a medium-

resolution infrared grating spectroradiometer and a diffraction grating disperses the incoming infrared radiation into 17 linear detector arrays comprising 2378 spectral samples. The spectral resolution of AIRS is about 0.5 cm$^{-1}$ and declined to 2 cm$^{-1}$ as the wavelengths are long and shorter respectively. It also measures trace gases such as $O_3$, CO and to some extent $CO_2$. AIRS $CH_4$ retrievals have been characterized and validated by Xiong et al. (2008).

## 5    Comparison of FTIR with MIPAS, MLS and AIRS observations

## 5.1    Comparison Methodology

Comparisons of daily average ground-based FTIR measurement of $CH_4$ and $N_2O$ with that of MIPAS were performed for time period of May, 2009 to December 2010. MIPAS, MLS and AIRS retrievals were used after averaging data obtained within coincident criteria of $\pm 2°$ of latitude and $\pm 10°$ of longitude from the ground based FTIR site in Addis Ababa and within time difference of $\pm 24$hr.

MIPAS version, V5R_CH4_224, V5R_N2O_224, MLS V3.3 and AIRS have better vertical resolution than ground-based FTIR profiles and high temporal and spatial coverage in the tropics. Hence, the profiles from MIPAS and MLS have been degraded to make a comparison between the FTIR and satellite observations. Therefore, the satellite measurement profiles are smoothed using the FTIR averaging kernels of individual species obtained from the ground based FTIR retrieval by applying the procedures reported in Rodgers and Connor (2003) and given as

25        $$\mathbf{x}_s = \mathbf{x}_a + \mathbf{A}(\mathbf{x}_i - \mathbf{x}_a) \qquad (3)$$

where $\mathbf{x}_s$ is the smoothed profile, $\mathbf{x}_a$ and $\mathbf{A}$ represents the a priori and averaging kernel for $CH_4$ and $N_2O$ obtained from the ground-based FTIR instrument respectively and $\mathbf{x}_i$ is the initial retrieved profile obtained from satellite measurements after we interpolated it to the FTIR grid spacing.

    We also calculate the following error parameters that can characterize the features of the instruments and the parameters to

be observed, such as the bias between the instruments using the difference (absolute or relative) of the daily mean profile. The





absolute or relative difference at each altitude layers of a pair profile is calculated using

$$\delta_i(z) = [\text{FTIR}_i(z) - \text{Sat}_i(z)] \tag{4}$$

The mean squares error can be expressed as

$$MSE_i(z) = \sqrt{\frac{1}{N(z)-1} \sum_{i=1}^{N(Z)} [\delta_i(z)]^2} \tag{5}$$

and the relative difference of a paired profile can be given in percentage as

$$\delta_i(z) = 100(\%) \times \frac{[\text{FTIR}_i(z) - \text{Sat}_i(z)]}{[\text{FTIR}_i(z) + \text{Sat}_i(z)]/2} \tag{6}$$

The mean difference (absolute or relative) for a complete set of coincident pairs of profiles obtained from the ground based FTIR and the correlative satellites is expressed as

$$\triangle_{\text{rel}}(z) = \frac{1}{N(z)} \sum_{i=1}^{N(z)} \delta_i(z) \tag{7}$$

where $\delta_i(z)$ is the difference (absolute or relative), $N(z)$ is the number of coincidences at $z$, $\text{FTIR}_i(z)$ is the FTIR VMR at $z$ and the corresponding $\text{Sat}_i(z)$ volume mixing ratio derived from satellite instruments. The standard deviation from the mean differences (absolute or relative) $\sigma_{diff}(z)$ is important to partially characterize the measurement error. As reported in von Clarmann (2006), some use de-biased standard deviation, which measures the combined precision of the instruments instead of the standard deviation of the mean differences.

$$\sigma(z) = \sqrt{\frac{1}{N(z)-1} \sum_{i=1}^{N(Z)} [\delta_i(z) - \triangle_{\text{abs}}(z)]^2} \tag{8}$$

where $\delta_i(z)$ is the difference (absolute or relative) for the $i^{th}$ coincident pair calculated using Eq.( 4) and Eq.( 6). The statistical uncertainty of the mean differences (absolute or relative), which is standard error of the mean (SEM) is the quantity used to judge the statistical significance of the estimated biases and it can be expressed in terms of the standard deviation of the mean:

$$SEM(Z) = \frac{\sigma(z)}{\sqrt{N(Z)}} \tag{9}$$

One can also conduct the comparison of FTIR and MIPAS using partial columns obtained from both FTIR and smoothed MIPAS $CH_4$ and $N_2O$. Hence, the relative difference between ground based FTIR and smoothed MIPAS partial columns of $CH_4$ and $N_2O$ by taking into account the lower altitude limit of MIPAS observations and upper limit of ground-based FTIR sensitivity has been calculated using

$$\text{RDiff}(\%) = 200 * (\frac{\text{PC}_{\text{FTIR}} - \text{PC}_{\text{Sat}}}{\text{PC}_{\text{FTIR}} + \text{PC}_{\text{Sat}}}) \tag{10}$$

where PC is partial column of FTIR and the corresponding satellite measurements.





Here in this paper coincidence and smoothing errors are not taken into account in the full error analysis during the comparisons between remotely sensed data sets in addition to the systematic and random uncertainties associated with each data set (von Clarmann, 2006). Hence, we will focus on the random uncertainties associated with both data sets in evaluating the uncertainties associated with the comparisons. However, the residual coincidence and horizontal smoothing errors are important

to elaborate the agreement between the data sets.

## 5.2    Comparison of FTIR $CH_4$

In Fig. 7 mean profiles, mean differences and estimated errors versus deviations of the difference between FTIR and MIPAS V5R_CH4_224 mixing ratios are shown.

The comparison has been made using 29 coincident data for a time period between Nov., 2009 and Dec., 2010. Middle
panel of Fig. 7 indicate a negative bias of -4.8% at around 16 km and 2% at 22 km. Between 23 and 27 km the FTIR value is higher than MIPAS values. The difference increases with altitude increases from 23 to 27 km (4.6%) with a maximum at 27 km. A large negative bias in FTIR $CH_4$ is obtained, i.e., FTIR $CH_4$ values are lower by 0.07 (4.8%) to 0.04 ppmv (2.2%). MIPAS V5R_CH4_222 profiles is biased high (14%) below 20-25 km as compared with other instruments Laeng et al. (2015) meanwhile the positive bias in the lowermost stratosphere and upper troposphere MIPAS-ENVISAT $CH_4$ and $N_2O$ profiles
version V5H_CH4_21 and V5H_N2O_21 and V5R_CH4_224, V5R_CH4_225, V5R_N2O_224 and V5R_N2O_225 products has been largely reduced (Plieninger et al., 2015, 2016).

Fig. 7 (right panel) indicates that the standard deviation of the mean differences is larger than the combined random error of the two instruments throughout the altitude. For instance, it is twice the combined standard deviation in the altitude above 20 km and less below 20 km, which indicates the underestimation of random errors of one or both of the instruments. In addition,
the overestimation of standard deviation of the difference may result from not taking all the error budget of MIPAS into account and the spatial and temporal criteria sets used to collect the coincidence data of MIPAS can create a discrepancy as well. The natural variation of the methane have also contributed to the overestimation of a standard deviation of the difference as biases vary with seasons as reported in Payan et al. (2009).

Fig. 8 (middle panel) shows the comparison between FTIR $CH_4$ profiles and $CH_4$ derived from MLS measurements of
atmospheric water vapor ($H_2O$), carbon monoxide (CO) and nitrous oxide ($N_2O$) and indicates that no significant bias in FTIR $CH_4$ data is present between 18 and 20 km. In the tropopause layer, the comparison indicates a negative bias of -1.7% at 17 km, i.e., the FTIR value is slightly high. FTIR $CH_4$ values are lower in altitude between 20-27 km with a bias of below 11% which is maximum at 27 km or on average by 0.12 ppmv (6.7%) between 20-27 km. The bias below 19 km and above 27 km can not be explained by the systematic errors of FTIR as the bias is larger than the systematic errors of FTIR and the later is
also out of the sensitivity ranges of FTIR. Furthermore, the standard deviation of the difference is larger than the combined random errors of the instruments. A bias in altitude range of 20 to 27 km can be explained by the systematic error of FTIR.

In Fig. 9 mean profiles, mean differences and estimated error versus deviation of the difference between FTIR and AIRS mixing ratios are shown. The largest negative bias is found in altitude between 11-19 km with a maximum difference of -0.08 ppmv at around 15 km. A negative bias that AIRS mixing ratio of $CH_4$ is higher than the FTIR as shown in Fig. 9. A positive





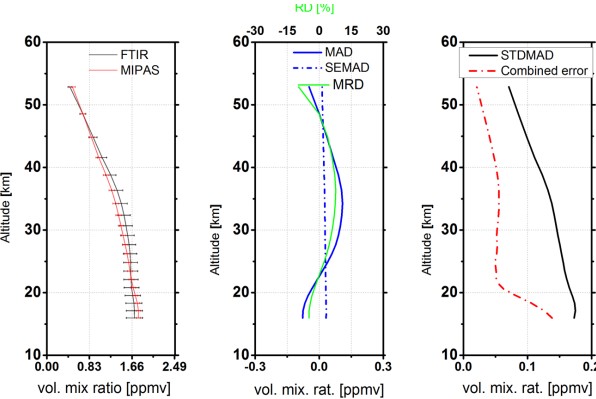

**Figure 7.** Comparison of $CH_4$ from MIPAS reduced resolution (V5R_CH4_224) and FTIR. Left panel: mean profiles of MIPAS (red) and FTIR (black ) and their standard deviation (horizontal bars). Middle panel: mean difference FTIR minus MIPAS (blue solid), standard error of the difference (blue dotted), mean relative differences FTIR minus MIPAS relative to their averaged (green, upper axis). Right panel: combined mean estimated statistical error of the difference (red dotted, contains MIPAS instrument noise error and FTIR random error budget), standard deviation of the difference (black solid).

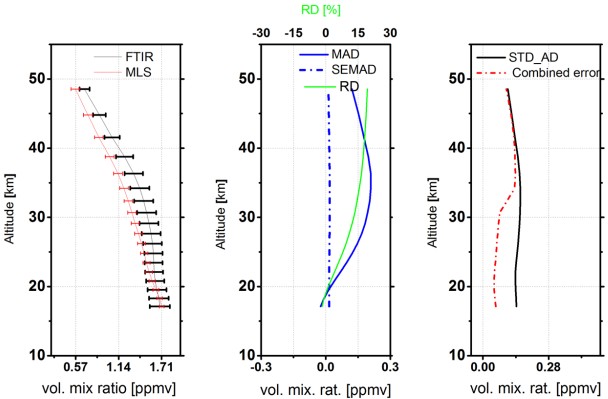

**Figure 8.** Comparison of $CH_4$ from MLS (V3.3) and FTIR. Details as in Fig. 7

bias existed at altitude between 7-9 km and similarly, it also shown in altitude between 21-27 km with a maximum value at around 27 km and its bias is 0.14 ppmv (9%). The standard deviation of the difference agrees to the combined random error in altitude below 20 km and it overestimate above 20 km.





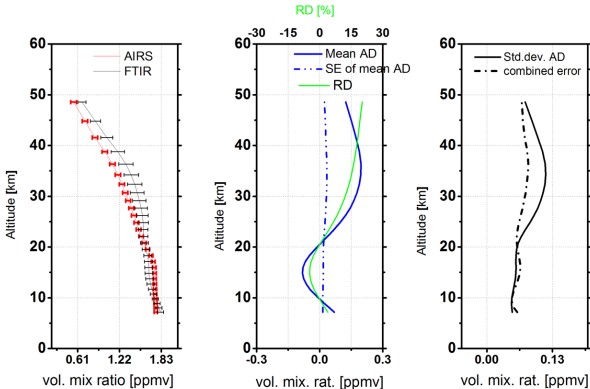

**Figure 9.** Comparison of $CH_4$ from AIRS and FTIR. Details as in Fig. 7

## 5.3 Comparison of FTIR $N_2O$

FTIR $N_2O$ mixing ratio comparison result is shown in Fig. 10, where it represents the mean profiles, mean absolute difference and standard deviation of the mean along with the combined errors of the two instruments. Mean profiles of FTIR show a maximum at around 23 km and decreases smoothly as altitude increases and that of MIPAS V5R_N2O_224 value starts to decline starting from the lowermost stratosphere.

Comparison of FTIR $N_2O$ profiles to MIPAS (V5R_N2O_224) measurements (see Fig. 10 (middle panel)) indicates that FTIR value is higher than the MIPAS above 20 km and the maximum mean absolute difference of $N_2O$ is 15% (0.04 ppmv) at around 24 km while, the FTIR value is less in altitude below 20 km with a maximum difference of -7% (-0.02) at around 17 km. The bias at 19 km is not statistically significant as the standard error of the mean is larger than the bias. In the remaining altitudes the standard error of the mean is smaller than the mean bias and the biases are statistically significant. Since, the bias in altitude between 20 to 27 km is smaller than the FTIR systematic errors, the bias can be explained in terms of systematic uncertainties in FTIR (see Fig. 5 (bottom middle panel)). The standard deviation of the difference is larger than the combined error of the two instruments in the altitude above 20 km (see Fig. 10, right panel) and the standard deviation of the difference agrees with the estimated combined random error in the altitude ranges between 20 to 27 km. For the altitudes below 20 km, the estimated combined random error is overestimated.

The left panel of Fig. 11 represents the mean profiles of $N_2O$ derived from the coincident pairs of FTIR and MLS $N_2O$. Throughout the whole altitude range, the value derived from FTIR is overestimated. The FTIR values of $N_2O$ are larger than the MLS value of $N_2O$ by a factor of 1.2 and 1.1 at around 21 and 27 km. The mean relative difference of FTIR and MLS $N_2O$ value increases as altitude increase, its value is less than 18.6% in altitudes below 27 km and its bias below 22 km is less than 8% that can be explained in terms of the systematic error of FTIR $N_2O$. The positive bias is statistically significant as the mean difference of the comparison is larger than the standard error of the mean.



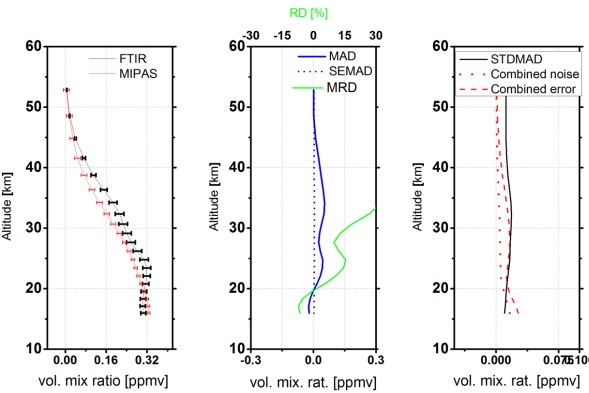

**Figure 10.** Comparison of $N_2O$ from MIPAS (V5R_N2O_224) and FTIR. Details as in Fig. 7

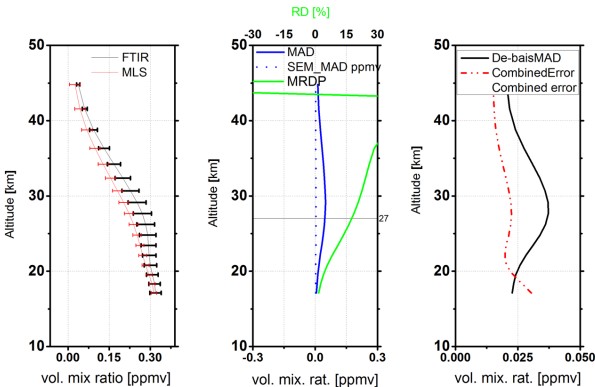

**Figure 11.** Comparison of $N_2O$ from MLS (V3.3) and FTIR. Details as in Fig. 7

For the partial column (PC) comparisons of FTIR with MIPAS, it is vital to take into account the lower altitude limit of MIPAS, which is 15 km for both target gases and the ground-based FTIR sensitivity is used to determine the upper altitude limit, which is reasonable up to $\sim$ 27 km for $CH_4$ and $N_2O$ in the tropical atmospheric condition. Therefore the PC that we use in the comparison is limited to the altitude ranges covered by both instruments. The DOFS within the partial columns limit are about 1.00 for $CH_4$ and about 1.2 for $N_2O$.

Fig. 12 shows the time series of the partial columns and relative differences of $CH_4$ (upper panel) and $N_2O$ (lower panel). The partial column comparison of $CH_4$ between values of FTIR and MIPAS revealed a mean error of -5.5%, mean squares error of 7.4% and a standard deviation from the mean error of 5%. Similarly, $N_2O$ values between FTIR and MIPAS revealed a





mean error of 0.5%, mean square error of 3.7% and standard deviation from mean error of 3.8%. in the latter case a significant positive bias is observed and in $CH_4$ negative bias was obtained.

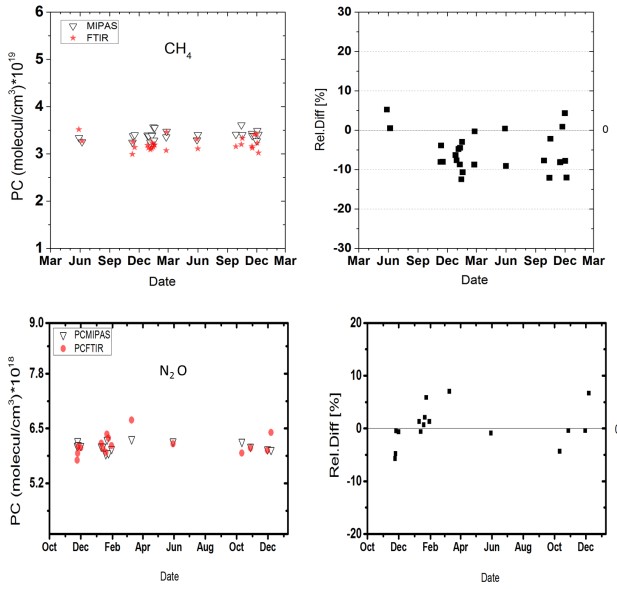

**Figure 12.** Time series of $CH_4$ and $N_2O$ partial column comparisons: right panel: ground based FTIR (stars) and MIPAS (V5R_CH4_224 and V5R_N2O_224) (triangular) partial columns. left panel: relative differences between ground based FTIR and MIPAS (V5R_CH4_224 and V5R_N2O_224) partial columns.

# 6   Summary and conclusions

The vertical profiles and partial columns of $CH_4$ and $N_2O$ over Addis Ababa, Ethiopia were derived from ground-based FTIR,
5   which is a very useful technique to derive vertical profiles and total column abundances of many important trace gases in the atmosphere. The mean partial column of $CH_4$ and $N_2O$ within the sensitivity ranges of the instrument, which is from the surface to around 27 km is determined as $2.85 \times 10^{19}$ molecules cm$^{-2} \pm 5.3\%$ and $5.16 \times 10^{18}$ molecules cm$^{-2} \pm 6.95\%$ respectively. The overall contribution of both statistical and systematic errors, i.e. total error of $CH_4$ and $N_2O$ from ground based FTIR is 3.1% and 3%, respectively.

10   The comparison of FTIR $CH_4$ and $N_2O$ with MIPAS IMK/IAA products of V5R_CH4_224 and V5R_N2O_224, version 3.3 MLS of $N_2O$ and $CH_4$ data and AIRS $CH_4$ are discussed in this paper. However, Version 3.3 MLS of $CH_4$ data were not directly derived from MLS, but the vertical profiles used in the study are derived from coincident measurements of atmospheric water vapour ($H_2O$), carbon monoxide (CO) and nitrous oxide ($N_2O$) by EOS MLS (Earth Observing System Microwave Limb Sounder) instrument on the NASA Aura satellite. From comparison of FTIR $CH_4$ and IMK/IAA MIPAS V5R_CH4_224 and





V5R_N2O_224 products, a statistically significant maximum negative bias of -4.8% in altitude 15 km that extends to 21 km and maximum positive bias of 4.6% in an altitude 27 km were obtained. The largest negative bias is found in an altitude between 11-19 km with a maximum difference of -0.08 ppmv (-4.8%) at around 15 km and a positive bias of less than 0.14 ppmv (9%) is found in altitude between 21-27 km with a maximum value at around 27 km in FTIR $CH_4$ comparison with

AIRS. On the other hand, comparison of $CH_4$ from ground based FTIR and MLS version 3.3 indicates a significant positive average bias of 0.12 ppmv (6.7%) in altitude range of 20-27 km and a negative bias -1.7% is also found at 17 km. Whereas in the case of FTIR and MIPAS V5R_N2O_224, a significant positive bias of less than 15% in the altitude range 22-27 km with a maximum value at around 25 km and a negative bias of -7% at 17 km has been obtained. A positive bias of less than 18.6% for the altitude below 27 km is noted for $N_2O$ between FTIR and MLS v3.3 and its bias below 22 km is less than 8% that can

be explained interms of the systematic error of FTIR $N_2O$. Moreover, the FTIR values of $N_2O$ is larger than MLS value by a factor of 1.2 and 1.1 at around 27 and 21 km, respectively. Therefore, the performance of instruments, FTIR, MIPAS and MLS in capturing $CH_4$ and $N_2O$ values at Addis Ababa station is good to study tropical atmospheric constituents. Comparison of the bias found for tropical atmospheric state with bias for other two atmospheric states (i.e. mid and high latitudes) will be the subject of the upcoming paper.

**acknowledgements**

. We are grateful to Goddard Space Flight Center and WACCM for providing temperature, pressure and a priori profiles of all molecules. Besides, AIRS and MLS data were obtained through the Goddard Earth Sciences Data and Information Services Center (http://daac.gsfc. nasa.gov/). We greatly acknowledge the MIPAS science teams for providing data used in this study. Finally, authors would like to thank Mekelle and Addis Ababa universities for the sponsorship and financial supports.





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
