# Peer review of "Methane and nitrous oxide from ground-based FTIR at Addis Ababa: observations, error analysis and comparison with satellite data."

_Atmospheric Measurement Techniques, 2019_

## Referee Comment (RC1) · Anonymous Referee #1 · 27 Jul 2019

This paper presents retrievals of CH4 and N2O using a ground-based FTIR instrument at Addis Ababa, Ethiopia with aim to present observations, error analysis, and comparisons with satellite data. The lack of long-term remote sensing observations at Addis Ababa makes this work important. The technique and results of such measurements might be interesting and likely suitable for the journal. However, I have major comments and foremost revisions are warranted before publication. In my opinion, the quality of the paper needs to be improved before publication.

Major Comments

[Figure]

I have the following major comments:

(1) It is not clear to me what exactly is (are) the goal(s) the authors try to achieve. There is a lack of description in both the FTIR measurements and comparison with satellites. The authors need to specify and emphasize what is(are) the goal(s). Is the goal to present a retrieval strategy of CH4 and N2O? or compare/validate three satellites with the ground-based FTIR?. In the manuscript, it is mentioned that a satellite is used to validate the FTIR, which is quite surprising. Normally, high-resolution FTIRs are used to validate satellite retrievals. In general, the manuscript is short and lack important details in many sections, e.g., FTIR measurements, satellite, and results.

(2) The retrieval strategy applied in this work is different than the NDACC/IRWG recommendations. I highly suggest to try the harmonized NDACC suggested retrievals and compare with your results. In particular, micro-windows applied here are different that NDACC recommended micro-windows. Furthermore, Sussman et al. (2011) found that HDO is an important interfering specie. However, here is not included or even mentioned.

(3) The DOFs obtained from FTIR measurements is limited to a value of approximately 2. Hence, the typical information content will allow to retrieve tropospheric and stratospheric columns. However, the authors show comparison of profiles with satellite, which are mainly sensitive in the stratosphere. Furthermore, comparisons are carried out using limited number of years, even though measurements at Addis Ababa started in 2009. Additionally, the criteria to establish coincident measurements between FTIR and satellite needs to be revised.

Specific Comments

I highly suggest to review exhaustively the English along the manuscript. I have some specific comments below, but they are not exhaustive by any means. P1, L2: Change Addis Ababa with Addis Ababa, Ethiopia

[Figure]

Is the instrument/site part of the NDACC effort?, if not please explain the reason.

Are the measurements automated?, how often do you measure.

Add information about quality control of spectra acquired.

Change ground based with ground-based when mentioned in the manuscript

P3 l13. Change "The Addis Ababa FTIR spectroscopy" for "The ground-based FTIR at The Addis"

P3, L15 add Altitude of the site

P.3, Section: In the measurement site section is not clear whether the site is located in the city limits of Addis Ababa or whether is located far from major emission sources. Please add information regarding typical air masses transported at Addis Ababa and/or emitted from local sources. Also, are there other atmospheric measurements carried out in Addis Ababa, which can be used to complement your study?.

P3, L23: remove very

P3, L23, change "to the study of trace gases in the atmosphere" with "to study trace gases in the atmosphere"

P3, L23, remove "terrestrial"

P3, L30: remove very

P4, L7: explain why Tikhonov-Phillips regularization was used and also why is the retrieval performed on a logarithmic scale.

P4, L29: The link http://www.www2.cesm.ucar.edu/working-groups does not work. Similarly, the link: http : //hyperion : gsf c : nasa : gov/Dataservices/automailer/index : html . Consider changing and being consistent with format.

P5, L1: Why different versions of HITRAN were used. Please explain and also, what versions were used for gases.
P5: I suggest to remove Figure 1. I do not see the value of Figure 1. It does not show a result/finding but only the a priori profiles used. Furthermore, in the text the temperature profile from this figure is not even mentioned. Keeping the text would be ok just change it accordingly.

P5. What do you mean by "The micro windows have been adopted from different sources.", do you mean past works? Expand a description and add references.

P5. L8-10. It is mentioned that micro windows are different than NDACC /IRWG guidelines. Please describe in detail why the NDACC guideline was not adopted. I was able to retrieve the guideline and all micro-windows for CH4 and N2O used in this work are different. I imagine the guidelines were created to obtain a harmonized retrieval strategy applied among different locations. This is important and needs to be explained.

In order to characterize the possible impact of the choice of the micro windows I suggest to test and compare the micro-windows/setting applied here with the harmonized NDACC settings.

Table 1. Recommendation: Change T.Gases with Gas; replace parenthesis with dash for micro-windows, change int. gases with interfering species; why is important to show three significant figures in DOF?, and mention here what species are retrieved as profiles and columns.

Table 1. According with the NDACC/IRWG guideline, HDO is an interfering specie but you are not using it, please explain. See also Sussmann et al. (2011).

Along the same lines, water vapor might influence the retrieval of CH4 and N2O. However, it is not mentioned what water vapor profile is used. Please describe if climatology (or reanalysis) is used, or do you pre-retrieve water vapor?.

P5, L9: change wondows with windows

P5, L10: The link www.ndacc.org does not have information regarding retrievals of CH4 and N2O. Change it accordingly.

Quality of Figures 2 and 3 is not good enough. They are blurry and too small to really see the quality of the fits.

P5, L11. I suggest to split the following sentence: The spectral fit and residual between measured and simulated spectra at five and four microwindows for CH4 and N2O respectively are depicted in Fig. 2 and Fig. 3 for example spectra recorded on Feb 26, 2013 and Dec 31, 2009 at Addis Ababa respectively." And why two different dates are used?, and remove Addid Adaba, you are not using other sites.

P5. L15: What do 0.4% and 0.35% mean?, explain.

P5, L15. It is mentioned that the retrieval strategy is optimized using a single spectra. . . please expand this description, what do you use a criteria for optimization? Is it consistent for all months, zenith angles?.

Regarding figure 4. Averaging kernel matrices are not described in the text. I recommend to remove the matrices and keep the rows of the averaging kernels. Rows of averaging kernels are not visible within the x-axis scale, please re-adjust. The sum of averaging kernels can be divided by 10 to use the same x-axis scale and lines need to be color coded by altitude and show the color bar. As all other figure, increase the quality of the figure.

P7, L12. Along the manuscript Addis Ababa is mentioned, although is clear. I suggest to remove Addis Ababa when is not needed.

P9, L10. Examples of error profiles are shown for CH4 - Feb 26 2013 and N2O – Dec 31 2009. Why was decided to use examples that are +3 years apart?

Figure 5. Please improve the quality of the figure. It is extremely hard to find the corresponding error type. I suggest to add the total error for each statistical and systematic errors. Additionally, I don't follow why the retrieved/apriori profiles are shown here. I suggest to replace this by the fraction of the total error with the retrieved profile and one might see the fractional error as a function of altitude.

P10, l7. It I mentioned that the MIPAS reduced spectral resolution is used, what does reduced resolution mean?

P10, L8. Explained why only satellite data between March 2009 to Dec 2010 is used for MIPAS?, what about the other satellite measurements (it is not mentioned)?

P11, L2. What does visibility flag 1 mean?, and how the criteria of diagonal elements has been chosen?

P11, L5. Change Plieninger et al. 2016 with Plieninger et al. (2016) and check format of references along the manuscript.

P11, L17. Under the MLS section it is mentioned: "In this work, we have used version 3.3 MLS of N2O data set to validate ground-based FTIR results". This is kind of surprising. Usually, the ground-based FTIRs are used to validate satellite-based measurements. Please explain in detail why you have chosen MLS to validate FTIR. Also, do you also use MIPAS and AIRS to validate FTIR?

P11, L19, Change EOS MLS (Earth Observing System) with EOS MLS

P12, L2. Expand the description "Selection criteria were implemented as stated in Livesey et al. (2013)". What do you mean by selection criteria?

In order to see the difference in sensitivity among the satellite measurements I suggest to include averaging kernels for the three selected satellites.

Section 5.1. The coincident criteria of $\pm 2$ deg latitude and $\pm$ 10 deg of longitude from the FTIR site is extremely large. As in other parts of the manuscript, please expand a description of why this wide range is used. I would try other distances as well. How do you assess the spatial-temporal variability of both CH4 and N2O.

Section 5.1. I encourage the authors to rename the following: V5R_CH4_224, V5R_N2O_224, MLS V3.3. These names are constantly but highly distractive.

P13, eq 4. It might be obvious but please describe variables of eq 4.

P13, eq 10. Why is this equation multiplied by 200?. Please revise this and all other equations.

Section 5.2. It is not explained why authors compare FTIR vs satellite vertical profiles. The FTIR information content is limited to 2 DOFs (tropospheric and stratospheric columns) but main figures for the comparison are shown as profiles.

References Sussmann, R., Forster, F., Rettinger, M., and Jones, N.: Strategy for high-accuracy-and-precision retrieval of atmospheric methane from the mid-infrared FTIR network, Atmos. Meas. Tech., 4, 1943-1964, https://doi.org/10.5194/amt-4-1943-2011, 2011.

―――――――――――――――――――――

---

## Referee Comment (RC2) · Anonymous Referee #2 · 28 Aug 2019

**Overview:**

Yirdaw berhe, et al., have submitted a manuscript comparing ground-based MIR-FTS measurements of atmospheric $CH_4$ and $N_2O$ at Addis Ababa, Ethiopia to that of three satellite (MIPAS, MLS and AIRS) data products. The manuscript details the Addis Ababa site, the measurements made and the spectral processing procedure (including retrieval uncertainty estimates). A brief overview of the satellite data products used are given, then coincident comparison criteria and lastly analysis of the profile and partial column comparison results.

The novelty of this manuscript is that this is the first time Addis Ababa FTIR $N_2O$ and $CH_4$ measurements are compared to satellite measurements.

The manuscript content is in the scope of the AMT journal. This research will be a welcome addition to already published literature concerning FTIR data from the Addis Ababa station, and also in the wider context of atmospheric ground-based trace gases measurements (including in situ) situated on the African continent (a data sparse region of the globe). Unfortunately, the manuscript is let down in multiple critical areas and I do not recommend publication until the issues listed below are addressed; either fixed or with a sufficient logical rebuttal.

**Specific comments:**

S1/ AMT English guidelines and house standards: A draw-back of the submitted manuscript is that I do not believe the grammar meets the standard required for publication in AMT. The authors are referred to AMT guidelines: https://www.atmospheric-measurement-techniques.net/for_authors/manuscript_preparation.html. There are instances of incorrect grammar use, ambiguous statements (most likely a consequence of improper grammar) and repetition of statements. All such instances need to be corrected. This is no reflection on the quality of the science presented and doesn't detract (only distracts and introduces ambiguity) from the novelty and importance of the presented subject matter (along with the effort the authors have already put into the manuscript). I would have expected the more experienced co-authors to have alerted the lead author to many of these grammatical and stylistic errors. For the manuscript review, correction of such grammatical errors will be left out (to speed up the review), and only commented upon if scientific clarity is required.

S2/ Could the authors clarify in the focus of the research. Comparisons are made between three satellite datasets and that of the ground-based FTIR measurements at Addis Ababa, but why? what is the motivation? In section 2.1 the manuscript alludes to why measurements at Addis Ababa are made, but only very broadly in a generic tropical atmosphere context. I gather the motivation is to use satellite measurements to validate the ground base measurements? This is unusual (usually the other way around), but a valid approach to help assess the quality of the ground-based measurements, if there is concern.

The authors state that the comparisons at the "Addis Ababa station is good to study tropical atmospheric processes" (Pg 19, L12). 'Good' in what context? Given the comparison results, will the ground-based $CH_4$ and $N_2O$ measurements capture seasonal cycles and multi-year trends? Will biomass burning or other episodic events most likely be seen, and from what part of the tropics (the tropics is a large place)?

S3/ Pg 3, L19. As, in S2, the authors give a generic/broad scale reason for the importance of trace gas measurements in the tropics. I recommend that a more specific reason/motivation for Addis

Ababa measurements be stated in the context of physical (or chemical) processes (emissions) related more specifically to Addis Ababa and the atmospheric footprint it 'sees'.

S4/ Pg 4, L3-24. Retrieval information is incomplete, see comment T31 below. After this sentence the authors start describing the retrieval specifications (spectral Microwindows and model atmosphere layer scheme), then return to describing the optimal estimation method (L8-L24). It would be better to complete describing the retrieval theory prior to specific retrieval strategies. The information supplied in L8-L24 is ubiquitous and generic, I do not think it needs description. This section could be condensed to a single sentence stating Roger OEM approach is used (referenced) with Tikhonov regularization(reference).

S5/ Pg4, L28: Apriori is mentioned. Are the apriori profiles used static? i.e. unvarying, or are they changing seasonally, yearly, or daily? If the apriori is static, then how is it constructed, a mean over XX years? Is the apriori based on a certain global region?

S6/ Pg 5, Fig 1. 'Tropics' is a big area with a variable atmospheric state. Do the authors mean the apriori over the Addis Ababa region? Could the date of the Apriori temperature profile be put in the figure caption? Also, to show the reader the variability of the atmospheric state, could the 1-sigma SD at each layer be plotted. The authors could also possibly omit figure 1 completely, as information content is minimal.

S7/ Pg 5, L15. The description "with positive and negative signs..." can be removed. This is implicit in the retrieval. The authors should also describe the residuals. Are they dominated by random or systematic uncertainties? For instance, in Fig 2, the $CH_4$ fit residuals are dominated by systematic spectral error, most likely due to imperfections in the spectroscopic database line parameters.

S8/ Pg 5, L15. The authors mention an "optimised retrieval strategy" but only give a passing mention to the Tikhonov retrieval regularization scheme. This is an important part of the retrieval; influencing overall information content and interlayer correlations of information content. Could the author please describe the Tikhonov regularization parameters. Why was the Tikhonov scheme implemented instead of using apriori uncertainties? What type of smoothing constraint is used (L1, L2 etc..), were the smoothing constraints normalised using layer thickness? what is the alpha parameter used? and how was the alpha parameter selected? is the alpha parameter static? or varies per retrieval?

S9/ Pg 7, L4. Since equation 1 may be eliminated in section 2.2, insert equation 1 here or a reference to this equation in Rodgers, 2000.

S10/ Pg 9, L21. Section 3.3 should end and new section 3.4 "Timeseries" (or something similarly named) should start. The content from L22 onwards (to end of the section) is concerned with the timeseries, not explicitly error estimation. The sentence starting "Concentrations of $CH_4$..." (Pg 6, L1) should be moved into this new section.

S11/ Figure 6 and Pg 6, L1. Please state the reason why is only data from 2009 to 2013 is analysed? I assume the Addis Ababa station is still currently (up to 2019) taking measurements?

S12/ Section 4: this section details MIPAS, MLS and AIRS satellite-based measurement platforms. It would be more helpful if the focus of this section was on details about Addis Ababa overpasses for each platform (such as the number of 'good' overpasses as a proportion of total). This would also help diagnose if Addis Ababa is a 'good' site (as the authors have stated) for such satellite validation.

S13/ Pg 11, L1. Why was only the period Mar2009 to Dec2010 used in MIPAS Addis Ababa comparisons? Why not longer?

S14/ Pg 12, L3. The last two sentences, starting with "Nitrous oxide derived…" should be omitted as it refers to MLS data version 2.2, not 3.3, unless the authors state (after verifying) that the precision of MLS N2O v3.3 is the same as v2.2.

S15/ Section 5 details and quantifies comparisons between the satellite data products and the Addis Ababa ground based FTIR data, but (in my opinion) does not elaborate on the results with respect to other ground based FTIR site measurements. Are the biases and spread seen at Addis Ababa like that of other ground-based FTIR sites (most likely also part of the NDACC)? This would help ascertain if the Addis Ababa is a 'good' validation site and is network comparable. All that is required is a literature review, this will help put the results derived in this study in context.

S16/ Equations 4 to 10 all pertain to statistical calculations between the FTIR and MIPAS measurements. I assume the same statistical methods are applied to comparisons with the other satellite data? Maybe make this section more generic, not just MIPAS specific.

S17/ Pg 14, L3. "Hence we will focus on the random uncertainties associated with…". This statement does not connect with the analysis in section 5.2. In section 5.2 dataset biases are quantified, which includes both random and systematic uncertainties (not separated). The standard deviations of the dataset comparisons will also include any systematic uncertainties. Maybe this sentence be retracted or changed to explain what is meant in a clearer manner.

S18/ Pg 14, L4. "However, the residual coincidence and horizontal smoothing errors…". If they are important why are they not investigated? The sentence starting on L19, pg14 ("In addition, the overestimation") also alludes (and offers conjecture) to issues arising around differences in the datasets relating to coincidence criteria but is not quantified. The authors could easily check this by changing the coincidence criteria (spatial and temporal) and see the effect of this in the dataset statistical differences.

S19/ Section 5.2 and Section 5.3. In both these sections there is no mention of how dataset degrees of freedom affect the profile differences. For example, the ground-based retrievals of $CH_4$ have approx. 2 DOFs. Differences at different altitudes will not be independent pieces of information. At Pg 14, L25 the authors state the bias of FTIR and MLS $CH_4$ at 18-20km is insignificant, at 17km -1.7%, and between 20-27km below 11%. Are these pieces of information independent? The authors may wish to comment on this fact and its implications.

S20/ Figures 10 and 11. There is no commentary on the large 'RD' differences above 30km (no sensitivity?). Could the authors comment on this?

S21/ Pg 17, L1. There needs to be a new section "section 5.4: Comparisons of partial columns" (or similar) starting at pg17 L1 if the authors are to start discussing partial column comparisons. Currently, the partial column comparisons for both $N_2O$ and $CH_4$ are under section 5.3.

S22/ Pg 17, L1. Why are only MIPAS partial column comparisons conducted? What about other satellite data products?

S23/ Fig 12. Uncertainty/error bars could be added to all data points. This would help in assessing the comparisons.

S24/ Pg 19, L12. Define 'good', do the authors mean the measurement quality and retrievals are 'good', or the location, or both? Since the focus of the manuscript is on assessing the performance of the Addis Ababa FTIR measurements, explaining 'good' is quite important.

**Technical comments (no particular order):**

T1/ Title: FTIR should be expanded, not an acronym. There is no need for the chemical formulas. The word 'measurements' should also be added after FTIR. So…I recommend the full title should be along the lines of: "Methane and nitrous oxide from ground-based Fourier transform infrared spectrometer measurements at Addis Ababa: observations, error analysis and comparison with satellite data."

T2/ Pg 1, L2. Possible change: "total column abundances and vertical distribution of various constituents in the atmosphere" to "total column trace gas abundances and vertical distributions".

T3/ Pg 1, L4. The superlative sentence "They reveal the high quality of FTIR measurements at Addis Ababa" is not required. The data and analysis reveal this.

T4/ In the abstract, I do not think it is necessary to specify satellite data product versions, for example 'V5R_CH4_224'. This is done in the main body.

T5/ Pg 1, L12. There are phases throughout the manuscript of the sort "a positive bias of less than 0.14 ppmv (9%) is found in the altitude range of 21 to 27 km". I gather this means there is a maximum positive bias of 0.14 ppmv in the range 21 to 27km? This may be a better way to state it.

T6/ Pg 2, L1. $CH_4$, $N_2O$ and CFCs are also stratospheric species…;)

T7/ Pg 2, L7. "ENIVSAT" to "ENVISAT satellite".

T8/ Pg 2, L9. Remove the word 'recent' from "The recent increasing…" (also replace 'to the' with 'on').

T9/ Pg 2, L10. Merge the sentences to read: "The recent increasing impact of $CH_4$ and $N_2O$ to the global warming has also been assessed by the last AR4 IPCC report (IPCC, 2007; Sussmann et al., 2012), additionally $N_2O$ will become the dominant ozone depleting substance emitted in the 21st century (Ravishankara et al., 2009)."

T10/ Pg 2, L11. What is IASI? Expand to say: "IASI instrument aboard the MetOp-2 satellite". MetOp-1 or MetOp-2…I can't remember.

T11/ Pg 2, L18. Rephrase first sentence: "In the tropics two important…"

T12/ Pg 2, L25. Replace 'launched' with 'taken'.

T13/ Pg 2, L27. Replace "The quality of ground based FTIR measurements" with "The Addis Ababa FTIR measurements".

T14/ Pg 2, L33. Replace 'confirm' with 'show'.

T15/ Pg 2, L34. Replace 'biased high and provided +14% as the most likely bias' with 'biased 14% high'.

T16/ Pg 2, L33. The reference Kenea throughout the manuscript should be replaced with Takele Kenea (2013)?

T17/ Pg 2, L33. The quoted references of Laeng (2015) and Plieninger (2016) refer to MIPAS comparisons with other satellite products. The paragraph starting at Pg 2, L25 concerns Addis Ababa FTIR measurements. There is a jump in topic. Reading as is, it could easily be taken that Addis Ababa measurements were used in these studies. These sentences should be removed or moved to a different part of the manuscript.

T18/ Pg 3, L3. The sentence "In this study, the previous work on intercomparison is extended to source gases $CH_4$ and $N_2O$ from ground-based FTIR" is quite ambiguous. Either remove or make more specific to Addis Ababa.

T19/ Pg 3, L7. "approach" can be replaced with 'strategy'.

T20/ Pg 3, L13. Is Addis Ababa part of the Network for the Detection of Atmospheric Composition Change (NDACC)? I suspect so, if this is the case it should be stated. The Takele-Kenea paper should be used as a site reference paper.

T21/ Pg 3, L15. Could the Addis Ababa site altitude (MASL) also be added?

T22/ Pg 3, L15. How is 'suitability' defined? Why is it suitable? Could possibly mention the amount of cloud free days a year.

T23/ Pg 3, L18. The superlative "extremely" can be removed, not needed.

T24/ Pg 3, L23. The superlative "very successfully" can be removed, not needed.

T25/ Pg 3, L24. Replace 'sun' with 'solar'

T26/ Pg 3, L27. The sentence "This technique…" should be moved to precede the sentence "The high resolution…"

T27/ Pg 3, L28. "Using seven narrowband filters". Assuming Addis Ababa is an NDACC site, do the seven filters meet NDACC specifications?

T28/ Pg 3, L29. It is mentioned an InSb detector is used to take measurements in over the range: 1500-4400cm$^{-1}$. There is no mention of detectors used to measure down to 750cm$^{-1}$, as mentioned in the prior sentence. Are measurements taken below 1500cm$^{-1}$?

T29/ Pg 3, L31. Replace "we used PROFFIT V…algorithm", with "we used the retrieval code PROFFIT (Ver95)".

T30/ Pg 4, L2. As the sentence reads, PROFFIT was developed to only retrieve $CH_4$ and $N_2O$. Could this sentence be corrected to reflect the fact PROFFIT was developed to retrieve multiple species.

T31/ Pg 4, L3. The sentence "This algorithm…" only tells half the information. Once a forward model calculation is completed, what happens next?

T32/ Pg 4, L4. At the end of the sentence "The vertical profiles…$N_2O$ respectively" change to "$N_2O$ respectively (see table 1 for spectral regions)." and could possibly be moved to section 3.1.

T33/, Pg4, L6. Could the bottom (base) layer height be stated.

T34/ Pg4, L27. replace 'setup' with 'strategy'

T35/ Pg 4, L29. This is the first time the 'NDACC' and 'IRWG' acronyms are used, please state in full.

T36/ Pg 5, L4. I think the sentence "The vertical variability…" is not required, or more information is required, i.e. define 'large'.

T37/ Pg 5, L8. "The micro-windows have been adopted from different sources". Why, and could the sources please be referenced.

T38/ Pg, 5, L9. Modified Microwindows: why is this?

T39/ Pg 5, L7. After the end of the sentence "lines are presented", the sentence Pg 4, L4 "The vertical profiles…" should be inserted.

T40/ Pg 5, L12. Remove the word "example". Also, could the authors detail the Signal to Noise ratio (SNR) of the Addis Ababa spectra. All this information is part of the 'optimization process'.

T41/ Table 1. could "int. Gases" be replaced with "Interfering gases". In the table legend, could "column amounts" be replaced with "total column amounts". Just an idea, (authors discretion), could another column be added for DOFs of the 0-27km partial column.

T42/ Figure 2. The usual convention is for measurement spectra data to be displayed as points (usually joined) and simulations as thin lines. The authors have the opposite of this. A minor point, and up to the authors if they would like to change it a more standard convention.

T43/ Figure 2 Legend. Acronyms are not explained prior, so the full name is required. Sorry. SEA is an unusual way to label/present the solar angle. Does SEA mean Solar elevation angle? The standard convention is solar zenith angle (SZA = 90.0 - SEA). I recommend that SZA be presented, not SEA. Spectra time is presented as "101715", please reformat to 10h17m15s or similar. Is SD the root mean square difference of the measured spectra and forward model? I.e. RMS. Please define.

T44/ Pg 7, L2. To be pedantic, "the most" can be replaced with "an" (as a matter of opinion).

T45/ Pg 7, L11. The authors should state at this point the units of the AVKs displayed, normalised to layer VMR or not? i.e. [VMR/VMR]

T46/ Pg 7, L16. Pedantic point, but any AVK that has non-zero elements has 'sensitivity', that is, infer information from the spectra. 0.5 is an arbitrarily defined 'cut-off'. So, I think a better statement would be "Fig 4 (top panel) shows that the…. has a sensitivity greater than 0.5 over the altitude range 2.45km to 27km".

T47/ Pg 8, L2. Could the altitude ranges of the two independent partial columns be stated.

T48/ Pg 8, L4. The sentence "The amplitude…" repeats information given earlier so can be removed.

T49/ Pg 8, L5. The sentence "We also ignore": Sorry I cannot understand this sentence (includes spelling mistakes), can it please be reworded and made clearer. The cited reference (Rinsland, 2005) does not contain any such information pertaining to altitude resolution. Can the authors please check the reference is correct or point out where it is in the paper.

T50/ Figure 4 legend. The datetime stamp of spectra the analysis is performed on should be given.

T51/ Pg 9, L4. Could the acronyms for sources of error be included, i.e. 'instrument line shape (ILS)', so that they correspond to the legend labels in fig 5.

T52/ Pg 9, L5. Can the term 'zero baselines offset' be 'zero level baseline offset'

T53/ Pg 9, L5. I assume statistical error means 'random' error, maybe rephrase as "statistical (random) error". Also take out "typical" as it relates directly to an example.

T54/ Pg 9, L17. Change part of the sentence "for the profile", to "for the $N_2O$ profile".

T55/ Pg 10, L6. The sentence starting "Vertical resolution…" needs to be referenced.

T56/ Pg 10, L8. The sentence starting "The analysis of the comparison…" should be moved to section 5.1.

T57/ Pg 11, L17. The starting sentence should be reworded: "…we have used the MLS $N_2O$ (v3.3) product to validate the ground-based Addis Ababa FTIR measurements."

T58/ Pg 11, L18. The altitude levels are given in pressure coordinates. So far in the manuscript, the altitude units have been in kilometres. Could the authors include the geometric altitude as well as the pressure. I.e. "100 and 0.1 hPa (XX to XX km's)"

T59/ Pg 12, L7. This sentence needs a reference.

T60/ Pg 12, L11. Unfortunately, the sentence starting "The spectral resolution…" does not make sense, could this be reworded, rephrased. The following sentence can be omitted as it does not add any information concerning this study.

T61/ Pg 12, L13. Could the data product version of AIRS $CH_4$ be added?

T62/ Pg 12, L20. The word 'version' can be removed.

T63/ Pg 12, L22. I think the meaning of 'degraded' means smoothed, so maybe the sentence could read "…MLS have been degraded (smoothed) to make a …" or replaced degraded with smoothed.

T64/ Pg 12, L29. Replace 'parameters' with 'statistics'.

T65/ Pg 13, L11. I assume that $Sat_i(z)$ is smoothed? So maybe state that: "and the corresponding Sati(z) smoothed volume mixing ratio is derived from…"

T66/ Figures 7, 8 and 9 could be combined into a single 3x3 figure, so could figures 10 and 11 (2x2)

T67/ Pg 14, L26. "In the tropopause layer", could the tropopause layer be defined, i.e. "In the tropopause layer (~XX-XX km)"

T68/ Figures 7,8,9,10 & 11. The legends in the figures are slightly different. Maybe standardise these as the figure captions all reference back to fig 7 caption. Figure 7 caption should also explain/define the legend captions. For example: "mean difference FTIR minus MIPAS (MAD, blue solid line) ..."

T69/ Pg 16, L2. The first sentence could be abbreviated to "FTIR $N_2O$ mixing ratio MIPAS comparison results are shown in fig 10."

T70/ Pg 16, L8. (-0.02 ppmv) ... include units please.

T71/ Pg 16, L11. Replace 'can' with 'could'.

T72/ Pg 16, L17. Add to end of sentence: "…the value derived from the FTIR is overestimated (relative to MLS)"

T73/ Pg 18, L4. The last part of the first sentence "…, which is a very useful…" is not needed.

T74/ Figure 12. The date label lacks information on the years(s), only months are given. Add information on the year(s) comparisons were made.

T74/ Acknowledgement: Remove full stop at the start of the first sentence. Remove the word 'besides'. Support, not supports.

T75/ References. The authors may wish to take out, or update references concerning an ACPD articles, such articles have not passed the peer review process. Reference formatting differs, so would be good to get it all consistent. Decide on an author convention for Samuel Takele Kenea, as this author is referenced a few times, but referenced differently (Takele Kenea S.).

---

## Author Comment (AC2) · 11 Oct 2019

I would like to thank the associate editor for handling my manuscript and the reviewer for their constructive comments on the manuscript. Here, the attached .pdf file is the responses to the referee comments. I hope, all the comments raised by the referee are addressed in the response and if there are points missed in the response, it will be taking into account in the revised manuscript.

Please also note the supplement to this comment:

[Figure]

https://www.atmos-meas-tech-discuss.net/amt-2019-170/amt-2019-170-AC2-supplement.pdf

---

## Author Response (AR1)

We would like to thank the associate editor for handling my manuscript and the reviewer for their constructive comments on the manuscript. Here, the attached .pdf file is the responses to the referee comments. I hope, all the major comments raised by the referee are addressed in the response and the technical comments have already corrected in the revised manuscript.

We thank the referee for very insightful questions and comments. They have helped to improve the quality of the paper. We have substantially revised and reorganized the manuscript, in many parts extended sentences and paragraphs have been added. Here, Our responses are given point-by-point below (blue Times New Roman font) following each of the reviewers' comments, which are repeated in full (black Times New Roman Italic font) mainly on the major comments. Reproduced text of the revised manuscript is set in green Times New Roman font in the responses. Repeated statements were deleted and stated in the response in red Times New Roman font in the responses. On the other hand, we showed the added texts and reproduced texts by highlighting (Green) in the revised manuscript. Finally, we have taken all the technical comments and English grammars on the revised manuscript.

The description on page 5, which discussed about Fig. 1 on the original manuscript has been deleted

Fig. 1 shows a priori profile of N2O and CH4 for tropical atmospheric conditions along with a temperature profile.

The following paragraph has been replaced by a statement as it has already been discussed in section 2.2.

"The averaging kernel is an important diagnostic tool to characterize to which degree the result represents measurement or a priori information by taking the sum of the individual elements of the rows of averaging kernels. Thus, **x,** which is the solution of retrieval as mathematically expressed in Eq.(1) is a combination of a priori profile $x_a$ and the differences of true values and a priori weighted by the averaging kernel matrix. Ideally the vertical resolution of the retrieval matches with the layer spacing used for the representation of state vector. In this case the average kernel would be the identity matrix. In reality, the diagonal values of the averaging kernel matrix are below unity, indicating that at a certain altitude the retrieved value represents either a priori information or that the value of atmospheric state is influenced by a state at neighboring altitudes. The vertical resolution is defined as the full width at half maximum (FWHM) of the rows of the averaging kernels."

The spectral resolution of a measurement affects the amount of vertical information derived from the spectral line shape of a measured species (Livesey et al. 2008).

The statements in p7, last paragraph have been deleted as it was stated in the introduction section.

"A comparison of MIPAS IMK/IAA product versions V5R_CH4_224 and V5R_N2O_224 with profiles measured by other instruments can be found in Plieninger et al. (2016). Laeng et al. 2015 had reported that MIPAS V5R_CH4_222 profiles are biased high (14%) below 20-25 km. The retrieval set up for the new MIPAS-ENVISAT CH4 and N2O profiles versions

V5R_CH4_224, V5R_CH4_225, V5R_N2O_224 and V5R_N2O_225 have been improved leading to reduced positive bias below 25 km with respect to other instruments (Plieninger et al., 2015, 2016)."

The statements on p8-9, the last paragraph has been deleted as it was stated in the introduction section.

"MIPAS V5R_CH4_222 profiles are biased high (14 %) below 20-25 km as compared with other instruments Laeng et al. (2015) meanwhile the positive bias in the lowermost stratosphere and upper troposphere MIPAS-ENVISAT CH4 and N2O profile version V5H_CH4_21 and V5H_N2O_21 and V5R_CH4_224, V5R_CH4_225, V5R_N2O_224 and V5R_N2O_225 products has been largely reduced (Plieninger et al., 2015, 2016)."

The following statements have been added in page 9 line 3 after the period that describes about the total fractional error of both CH4 and N2O.

The total fractional error of CH4 and N2O retrieved from ground-based FTIR has been shown in the last column of Fig. 4. Fractional error of CH4 is less than 10 % in the altitude below 27 km with minimum fractional error of 4 at middle troposphere. On the other hand, the total fraction error of N2O retrieval is less than 13 % in the altitude below 27 km with a minimum value of 4 %  at 6 km and 7.5 % at 17 km

**Responses to referee 1: (received: 27 July 2019)**

*This paper presents retrievals of CH₄ and N₂O using a ground-based FTIR instrument at Addis Ababa, Ethiopia with aim to present observations, error analysis, and comparisons with satellite data. The lack of long-term remote sensing observations at Addis Ababa makes this work important. The technique and results of such measurements might be interesting and likely suitable for the journal. However, I have major comments and foremost revisions are warranted before publication. In my opinion, the quality of the paper needs to be improved before publication.*

Response: We would like to thank the reviewer for this positive evaluation and critical comments that would help us to make the paper more vital to the scientific community. Furthermore, the response of each major comments and specific comments will place following each comment.

Here, some of the figures we have shown in this paper are important as the article is the first result of FTIR CH4 and N2O, such figures would be ignored while we prepared other related works. Some responses explained below are related to the work done here is the first for this site.

**Major Comments**

*I have the following major comments:*

> *(1) It is not clear to me what exactly is (are) the goal(s) the authors try to achieve. There is a lack of description in both the FTIR measurements and comparison with satellites. The authors need to specify and emphasize what is (are) the goal(s). Is the goal to present a retrieval strategy of CH₄ and N₂O? or compare/validate three satellites with the ground-based FTIR?. In the manuscript, it is mentioned that a satellite is used to validate the FTIR, which is quite surprising. Normally, high-resolution FTIRs are used to validate satellite retrievals. In general, the manuscript is short and lack important details in many sections, e.g., FTIR measurements, satellite, and results.*

Response: The goals of this paper are retrieval strategy of CH₄ and N₂O, since the micro windows applied here are somewhat modified. Furthermore, validations of the ground based FTIR CH4 and N2O have been made. As the referee has stated, FTIR is a high resolution and used to validate the satellite observations. However, the FTIR at Addis Ababa is new and it required to verify the measured parameters through validation.

> *(2) The retrieval strategy applied in this work is different than the NDACC/IRWG recommendations. I highly suggest to try the harmonized NDACC suggested retrievals and compare with your results. In particular, micro-windows applied here are different that NDACC recommended micro-windows. Furthermore, Sussman et al.*

*(2011) found that HDO is important interfering specie. However, here is not included or even mentioned.*

Response: The reasons why we modify the micro windows are due to high residuals obtained between the measured and synthesized spectra at the Addis Ababa site. Thus, the micro-windows recommended by the NDACC might be useful for the other FTIR sites found at mid and high latitudes. The Micro windows applied here are different from the micro windows used by Sussman et al. (2011) and that is why the HDO is not an important interfering species.

*(3) The DOFs obtained from FTIR measurements is limited to a value of approximately 2. Hence, the typical information content will allow to retrieve tropospheric and stratospheric columns. However, the authors show a comparison of profiles with satellite, which are mainly sensitive in the stratosphere. Furthermore, comparisons are carried out using a limited number of years, even though measurements at Addis Ababa started in 2009. Additionally, the criteria to establish coincident measurements between FTIR and satellite needs to be revised.*

Response: As the referee stated, the DOFs obtained from FTIR measurements are 2. Due to the limitation of the sensitivities of satellite that is upper troposphere and stratosphere, the comparisons have been done on the stratosphere to increase the number of coincident days. During the measurements of FTIR, there are several days and months where the instruments were not functional. To make it more clear the period time that does not have a measured value will be stated (added) as a new description in section 2.1.

***Specific Comments***

*I highly suggest to review exhaustively the English along with the manuscript. I have some specific comments below, but they are not exhaustive by any means. P1, L2: Change Addis Ababa with Addis Ababa, Ethiopia*

Response: Done

*Is the instrument/site part of the NDACC effort?, if not please explain the reason.*

Response: The instrument or site is not part of NDACC yet. Since all the measured species have not yet validated, only ozone and water vapour are validated by the previous PhD student (Dr. samual T.). This article was prepared to register the site as members of NDACC and other works has also submitted.

*Are the measurements automated?, how often do you measure.*

Response: No, we measured them starting from May 2009 to February 2013 with some days and months are missing (See Section 2.1).

*Add information about quality control of spectra acquired.*

Response: We have added the following sentence about the quality of the acquired spectra at the end of Section 2.2.

The quality of the measurements during the time period of May 2009-Feberoury 2011 has revealed by Takele Kenea et al., (2013).

*Change ground based with ground-based when mentioned in the manuscript*

Response: Done

*P3 l13. Change "The Addis Ababa FTIR spectroscopy" for "The ground-based FTIR at The Addis"*

Response on P3 L13: Done

*P3, L15 add Altitude of the site*

Response on P3 L15:  We added ", 2443 m a.m.l." after 38.76°E.

*P.3, Section: In the measurement site section is not clear whether the site is located in the city limits of Addis Ababa or whether is located far from major emission sources. Please add information regarding typical air masses transported at Addis Ababa and/or emitted from local sources. Also, are there other atmospheric measurements carried out in Addis Ababa, which can be used to complement your study?.*

Response on P3, Section 2.1:  As suggested by the referee, we have added sentences that describe the details of the measurement site, Addis Ababa and the period time when the instrument was operational. As far as I know, no other atmospheric measurements are carried on in Addis Ababa.

*P3, L23: remove very*

Response on P3, L23: Done

*P3, L23, change "to the study of trace gases in the atmosphere" with "to study trace gases in the atmosphere"*

Response on P3, L23: Done

*P3, L23, remove "terrestrial"*

Response on P3, L23: Done

*P3, L30: remove very*

Response on P3, L30: Done

*P4, L7: explain why Tikhonov-Phillips regularization was used and also why is the retrieval performed on a logarithmic scale.*

Response on P4, L7: Since the distribution of the a priori profile of the species has a logarithmic or exponential distribution in the upper troposphere and lower stratosphere over Addis Ababa.

*P4, L29: The link http://www.www2.cesm.ucar.edu/working-groups does not work. Similarly, the link: http : //hyperion : gsfc : nasa : gov/Dataservices/automailer/index : html . Consider changing and being consistent with format.*

Response on P4, L29: Changed to http://www2.cesm.ucar.edu/working_groups/?ref=nav and https://hyperion.gsfc.nasa.gov/

*P5, L1: Why different versions of HITRAN were used. Please explain and also, what versions were used for gases*

Response on P5, L1: As the referee suggested, we added sentences that description which versions of HITRAN data were used for gases. Different versions of HITRAN were used as the new updated HITRAN…

The updated HITRAN data of 2009 for $H_2O$ and HITRAN 2012 for $CO_2$, $CH_4$, NO2 and hit08 of $N_2O$ were used during retrieval of $CH_4$ and $N_2O$.

*P5: I suggest to remove Figure 1. I do not see the value of Figure 1. It does not show a result/finding but only the a priori profiles used. Furthermore, in the text the temperature profile from this figure is not even mentioned. Keeping the text would be ok just change it accordingly.*

Response on P5: We agree and removed it. Similarly, the text has been changed as follows.

Both methane ($CH_4$) and nitrous oxide ($N_2O$) are well-mixed in the troposphere and their VMR decrease with height and becomes negligible with no variation above 55 km. The vertical variability of $N_2O$ and $CH_4$ in the lower stratosphere is characterized by a large vertical gradient

*P5. What do you mean by "The micro windows have been adopted from different sources.", do you mean past works? Expand a description and add references.*

Response on P5: We mean, micro windows that give us a minimum residuals and errors were considered. The references from where the micro windows are adopted has been added at P5, L8.

The microwindows have been adopted from different sources (.Senten et al. 2008; Sussmann et al., 2011; Arndt et al., 2004).

*P5. L8-10. It is mentioned that micro windows are different than NDACC /IRWG guide-lines. Please describe in detail why the NDACC guideline was not adopted. I was able to retrieve the guideline and all micro-windows for $CH_4$ and $N_2O$ used in this work are different. I imagine the guidelines were created to obtain a harmonized retrieval strategy applied among different locations. This is important and needs to be explained.*

Response on P5, L8-10: The reasons why we modify the micro windows are due to high residuals obtained between the measured and synthesized spectra at the Addis Ababa site. Thus, the micro-windows recommended by the NDACC might be useful for the other FTIR sites found in mid and high latitudes.

*In order to characterize the possible impact of the choice of the micro windows I suggest to test and compare the micro-windows/setting applied here with the harmonized NDACC settings.*

Response: We have done using the harmonized NDACC settings and comparing them with those used in this paper. However, the comparisons are not stated in the paper.

*Table 1. Recommendation: Change T.Gases with Gas; replace parenthesis with dash for micro-windows, change int. gases with interfering species; why is important to show three significant figures in DOF?, and mention here what species are retrieved as profiles and columns.*

Response on Table 1: As suggested by the referee, we agree that the title of the Table has to be clear; T. Gases have been changed to "Gas", MW to "micro-windows", int. gases to "interfering species".

*Table 1. According with the NDACC/IRWG guideline, HDO is an interfering specie but you are not using it, please explain. See also Sussmann et al. (2011).*

Response: As we have stated previously, the micro-windows used in this paper are different from the micro-windows recommended in the NDACC / IRWG guideline that have been applied to other atmospheric conditions.

*Along the same lines, water vapor might influence the retrieval of $CH_4$ and $N_2O$. However, it is not mentioned what water vapor profile is used. Please describe if climatology (or reanalysis) is used, or do you pre-retrieve water vapor?.*

Response on P5, Table 1: No, We use WACCM reanalysis data

*P5, L9: change wondows with windows*

Response on P5, L9: Done

*P5, L10: The link www.ndacc.org does not have information regarding retrievals of $CH_4$ and $N_2O$. Change it accordingly.*

Response on P5, L10: We have changed it adding the following instead of www.ndacc.org "the EU projects UTFIR (www.nilu.no/uftir) and HYMN (www.knmi.nl/samenw/hymn) reports."

*Quality of Figures 2 and 3 is not good enough. They are blurry and too small to really see the quality of the fits.*

Response on Figures 2 and 3: Those two figures have been updated as figure 1 and figure 2 (see on the revised manuscript page 6 and 7)

*P5, L11. I suggest to split the following sentence: The spectral fit and residual between measured and simulated spectra at five and four microwindows for $CH_4$ and $N_2O$ respectively are depicted in Fig. 2 and Fig. 3 for example spectra recorded on Feb 26, 2013 and Dec 31, 2009 at Addis Ababa respectively." And why two different dates are used?, and remove Addid Adaba, you are not using other sites.*

Response on P5, L11: Since we prepared the retrieval and validation results of both $CH_4$ and $N_2O$ separately. The sample spectra presented in the paper for CH4 and N2O were different. The sentence has also separated into two sentences.

The spectral fit and residual between measured and simulated spectra at five micro windows for $CH_4$ is shown in Fig. 2 for spectra recorded on Feb. 26, 2013. Whereas, four micro windows are used for $N_2O$ and depicted in Fig. 3 for spectra recorded on Dec 31, 2009.

*P5. L15: What do 0.4% and 0.35% mean?, explain.*

Response on P5, L15: To make clear why we have put those results in the paper, the following paragraph would be added.

Generally residual of the spectra mean that the difference between measured and synthesis spectra. Furthermore, the residual can also be expressed in percentage while we took the ratio of the difference and measured times 100. This residual was used to explain the quality of the measured spectra which we have used to derive the concentration or amount of both $CH_4$ and $N_2O$. The magnitude of residuals indicates that measured spectra which we have used to derive the concentration or amount of both CH4 and N2O.was quality as they are less than 1.

*P5, L15. It is mentioned that the retrieval strategy is optimized using a single spectra: : : please expand this description, what do you use a criteria for optimization? Is it consistent for all months, zenith angles?.*

Response on P5, L15: The optimization of the retrieval strategy starts from the selection of the micro windows which are somewhat different from the recommended by NDACC.

*Regarding figure 4. Averaging kernel matrices are not described in the text. I recommend to remove the matrices and keep the rows of the averaging kernels. Rows of averaging kernels are not visible within the x-axis scale, please re-adjust. The sum of averaging kernels can be divided by 10 to use the same x-axis scale and lines need to be color coded by altitude and show the color bar. As all other figure, increase the quality of the figure.*

Response Regarding figure 4: (see page 28)

*P7, L12. Along the manuscript Addis Ababa is mentioned, although is clear. I suggest to remove Addis Ababa when is not needed.*

Response on P7, L12: The Addis Ababa has been removed from the following specific locations in the text.

*P9, L10. Examples of error profiles are shown for CH4 - Feb 26 2013 and N2O – Dec 31 2009. Why was decided to use examples that are +3 years apart?*

Response on P9, L10: Because we first prepared all the analysis separately for $CH_4$ and $N_2O$

*Figure 5. Please improve the quality of the figure. It is extremely hard to find the corresponding error type. I suggest to add the total error for each statistical and systematic errors. Additionally, I don't follow why the retrieved/apriori profiles are shown here. I suggest to replace this by the fraction of the total error with the retrieved profile and one might see the fractional error as a function of altitude.*

Response on Figure 5.: We have improved the quality of the figure according to the suggestion of the referee and we added the following expressions about the fraction total error. (see revised manuscript on page 9)

*P10, l7. It I mentioned that the MIPAS reduced spectral resolution is used, what does reduced resolution mean?*

Response on P10, L7: The measurements collected between January 2005 and April 2012 it measured with a reduced spectral resolution (RR, theoretical resolution: 0.0625 cm$^{-1}$, apodised resolution: 0.121 cm$^{-1}$) but with a finer tangent altitude spacing. The data used in this paper is

*P10, L8. Explained why only satellite data between March 2009 to Dec 2010 is used for MIPAS?, what about the other satellite measurements (it is not mentioned)? why this wide range is used. I would try other distances as well. How do you assess the spatial-temporal variability of both CH4 and N2O.*

Response on P10, L8: We have added the statement to explain the period time used to validate FTIR with MLS. Here, in this paper, we did not put anything about the spatial-temporal variability of both $CH_4$ and $N_2O$.

The comparison of FTIR with MLS for a period time on May 2009 to February 2013 has also made.

*P11, L2. What does visibility flag 1 mean?, and how the criteria of diagonal elements has been chosen?*

Response on P11, L2: The visibility flag indicates whether spectral data was available for the given altitude (value=1) or not (value=0). Altitudes with visibility flag = 0 have to be omitted. In this paper, we have taken data with visibility flag value of 1. Similarly, the diagonal element of the averaging kernels that indicates the sensitivity of the instrument and its value above 0.03 has been taking on this work.

*P11, L5. Change Plieninger et al. 2016 with Plieninger et al. (2016) and check format of references along the manuscript.*

Response on P11, L5:  Done

*P11, L17. Under the MLS section it is mentioned: "In this work, we have used version 3.3 MLS of N2O data set to validate ground-based FTIR results". This is kind of surprising. Usually, the ground-based FTIRs are used to validate satellite-based measurements. Please explain in detail why you have chosen MLS to validate FTIR. Also, do you also use MIPAS and AIRS to validate FTIR?*

Response on P11, L17:  The results obtained from the ground-based FTIR observations at Addis Ababa are presented and discussed in this paper for the first time to this latitude band. This is the reason why we verify the quality of the FTIR measurements by MIPAS, MLS and AIRS.

*P11, L19, Change EOS MLS (Earth Observing System) with EOS MLS*

Response on P11, L19:  Done

*P12, L2. Expand the description "Selection criteria were implemented as stated in Livesey et al. (2013)". What do you mean by selection criteria?*

Response on P12, L2: The selection criteria mean that the status of the data sets concider in this work.

*In order to see the difference in sensitivity among the satellite measurements I suggest to include averaging kernels for the three selected satellites.*

*Section 5.1. The coincident criteria of 2 deg latitude and 10 deg of longitude from the FTIR site is extremely large. As in other parts of the manuscript, please expand a description of*

*Section 5.1. I encourage the authors to rename the following: V5R_CH4_224, V5R_N2O_224, MLS V3.3. These names are constantly but highly distractive. P13, eq 4. It might be obvious but please describe variables of eq 4.*

Response, Section 5.1.:  The paragraph and equations have been rewritten (see p19 )

*P13, eq 10. Why is this equation multiplied by 200?. Please revise this and all other equations.*

Response on P13, eq.10:  We multiply it by 200 because the denominator was multiplied by 0.5 and we corrected it (see p19 ).

*Section 5.2. It is not explained why authors compare FTIR vs satellite vertical profiles. The FTIR information content is limited to 2 DOFs (tropospheric and stratospheric columns) but main figures for the comparison are shown as profiles.*

Response, Section 5.2.:  Since those results are the first in Addis Ababa.

*References Sussmann, R., Forster, F., Rettinger, M., and Jones, N.: Strategy for high-accuracy-and-precision retrieval of atmospheric methane from the mid-infrared FTIR network, Atmos. Meas. Tech., 4, 1943-1964, https://doi.org/10.5194/amt-4-1943-2011, 2011.*

Response: corrected

**Responses to referee 2:   (received: 28 Aug 2019 )**

*Overview:*

*Yirdaw berthed, et al., have submitted a manuscript comparing ground-based MIR-FTS measurements of atmospheric $CH_4$ and $N_2O$ at Addis Ababa, Ethiopia to that of three satellite (MIPAS, MLS and AIRS) data products. The manuscript details the Addis Ababa site, the measurements made and the spectral processing procedure (including retrieval uncertainty estimates). A brief overview of the satellite data products used are given, then coincident comparison criteria and lastly analysis of the profile and partial column comparison results.*

*The novelty of this manuscript is that this is the first time Addis Ababa FTIR $N_2O$ and $CH_4$ measurements are compared to satellite measurements.*

*The manuscript content is in the scope of the AMT journal. This research will be a welcome addition to already published literature concerning FTIR data from the Addis Ababa station, and also in the wider context of atmospheric ground-based trace gases measurements (including in situ) situated on the African continent (a data sparse region of the  globe). Unfortunately, the manuscript is let down in multiple critical areas and I do not recommend publication until the issues listed below are addressed; either fixed or with a sufficient logical rebuttal.*

Response: We would like to thank the reviewer for this positive evaluation and critical comments that would help us to make the paper more vital to the scientific community. Furthermore, the response of each major comments and specific comments will place following each comment.  I hope, all the responses given in the manuscript have satisfied the referee.

*Specific comments:*

*S1/ AMT English guidelines and house standards: A draw-back of the submitted manuscript is that I do not believe the grammar meets the standard required for publication in AMT. The authors are referred to AMT guidelines: https://www.atmospheric-measurement-techniques.net/for_authors/manuscript_preparation.html. There are instances of incorrect grammar use, ambiguous statements (most likely a consequence of improper grammar) and repetition of statements. All such instances need to be corrected. This is no reflection on the quality of the science presented and doesn't detract (only distracts and introduces ambiguity) from the novelty and importance of the presented subject matter (along with the effort the*

authors have already put into the manuscript). I would have expected the more experienced co-authors to have alerted the lead author to many of these grammatical and stylistic errors. For the manuscript review, correction of such grammatical errors will be left out (to speed up the review), and only commented upon if scientific clarity is required.

Response on S1: We appreciated the referee for taking out all the concerns on the manuscript. Hopefully, all the issues raised above will be addressed in the revised manuscript.

The grammer errors have been corrected in the revised manuscript.

S2/ Could the authors clarify in the focus of the research. Comparisons are made between three satellite datasets and that of the ground-based FTIR measurements at Addis Ababa, but why? What is the motivation? In section 2.1 the manuscript alludes to why measurements at Addis Ababa are made, but only very broadly in a generic tropical atmosphere context. I gather the motivation is to use satellite measurements to validate the ground base measurements? This is unusual (usually the other way around), but a valid approach to help assess the quality of the ground-based measurements, if there is concern.

The authors state that the comparisons at the "Addis Ababa station is good to study tropical atmospheric processes" (Pg 19, L12). 'Good' in what context? Given the comparison results, will the ground-based $CH_4$ and $N_2O$ measurements capture seasonal cycles and multi-year trends? Will biomass burning or other episodic events most likely be seen, and from what part of the tropics (the tropic is a large place)?

Response on S2: Since this result is the first to the Addis Ababa site, it required validation to assess the quality of the ground-based FTIR measurements. The motivation is to fill the gap in understanding the atmosphere over Addis Ababa, tropical region of the globe. Moreover, there are scores of measurements in tropical. The expression below has been added in section 2.1 to elaborate more about the measurement site.

Thus, the observed variation in the measurement of atmospheric trace gases would help us to understand the effects of tropical dynamics on the site. Besides, it fills gap to the scarcity of ground based measurements in tropical.

The sentence at pg 19, L1, "Therefore, the performance of instruments, FTIR, MIPAS and MLS in capturing CH4 and N2O values at the Addis Ababa station is good to study tropical atmospheric constituents." has been rewritten as follows. Since, the paper concerns on the FTIR measurement (see next response S3)

S3/ Pg 3, L19. As, in S2, the authors give a generic/broad scale reason for the importance of trace gas measurements in the tropics. I recommend that a more specific reason/motivation for Addis Ababa measurements be stated in the context of physical (or chemical) processes (emissions) related more specifically to Addis Ababa and the atmospheric footprint it 'sees'.

Response on S3/pg3, L19: As the number of the population has increased dramatically and there is no encouraging methods for waste disposal at the Addis Abba. The following expression has been added in the manuscript as a description of the site.

Thus, the observed variation in the measurement of atmospheric trace gases would help us to understand the effects of tropical dynamics on the site. Besides, it filled the gap to the scarcity of ground based measurements in tropical.

*S4/ Pg 4, L3-24. Retrieval information is incomplete, see comment T31 below. After this sentence the authors start describing the retrieval specifications (spectral Microwindows and model atmosphere layer scheme), then return to describing the optimal estimation method (L8-L24). It would be better to complete describing the retrieval theory prior to specific retrieval strategies. The information supplied in L8-L24 is ubiquitous and generic, I do not think it needs description. This section could be condensed to a single sentence stating Roger OEM approach is used (referenced) with Tikhonov regularization (reference).*

Response on S4/page 4, L3-24: I think there are redundancy of concepts and we have deleted the sentences in L3-L9.

*S5/ Pg4, L28: Apriori is mentioned. Are the apriori profiles used static? i.e. unvarying, or are they changing seasonally, yearly, or daily? If the apriori is static, then how is it constructed, a mean over XX years? Is the apriori based on a certain global region?*

Response on S5/pg 4, L28: It was static, mean of 40 years WACCM for tropics was used.

*S6/ Pg 5, Fig 1. 'Tropics' is a big area with a variable atmospheric state. Do the authors mean the apriori over the Addis Ababa region? Could the date of the Apriori temperature profile be put in the figure caption? Also, to show the reader the variability of the atmospheric state, could the 1-sigma SD at each layer be plotted. The authors could also possibly omit figure 1 completely, as information content is minimal.*

Response on S6/pg 5, Fig 1: We agree and removed it. Similarly, the text has been changed as follows.

Both methane ($CH_4$) and nitrous oxide ($N_2O$) are well-mixed in the troposphere and their VMR decrease with height and becomes negligible with no variation above 55 km. The vertical variability of $N_2O$ and $CH_4$ in the lower stratosphere is characterized by a large vertical gradient

*S7/ Pg 5, L15. The description "with positive and negative signs..." can be removed. This is implicit in the retrieval. The authors should also describe the residuals. Are they dominated by random or systematic uncertainties? For instance, in Fig 2, the CH4 fit residuals are dominated by systematic spectral error, most likely due to imperfections in the spectroscopic database line parameters.*

Response on S7/pg 5, L15: The sentences "The magnitude of residuals of spectral fits are less than 1% with both positive and negative signs (CH4: 0.4%; N2O: 0.34%)." has been

replaced by "The magnitude of the residuals of spectral fits span a range of a maximum of +0.2 % to -0.64 % for CH4 and + 0.34 % to -0.34 % for N2O."

As clearly seen in the error analysis section of the manuscript pg 9 L9-21, the $N_2O$ fit residuals are dominated by statistical error.

*S8/ Pg 5, L15. The authors mention an "optimised retrieval strategy" but only give a passing mention to the Tikhonov retrieval regularization scheme. This is an important part of the retrieval; influencing overall information content and interlayer correlations of information content. Could the author please describe the Tikhonov regularization parameters. Why was the Tikhonov scheme implemented instead of using apriori uncertainties? What type of smoothing constraint is used (L1, L2 etc..), were the smoothing constraints normalised using layer thickness? what is the alpha parameter used? and how was the alpha parameter selected? is the alpha parameter static? or varies per retrieval?*

Response On s8/page 5, L15: detailed description of retrieval strategy has been added after L10.

Methane and nitrous oxide vertical profiles over Addis Ababa have been obtained by fitting five and four micro windows respectively. The retrieved state vector contains the retrieved volume mixing ratios of the target gas defined in 41 layers of the tropical atmospheric conditions. The retrieved profiles were derived using a Tikhonov-Phillips method on a logarithmic scale.

*S9/ Pg 7, L4. Since equation 1 may be eliminated in section 2.2, insert equation 1 here or a reference to this equation in Rodgers, 2000.*

Response on S9/pg 7, L4: The retrieved profile is a combination of a priori profile $x_a$ and the differences of true Values and a priori weighted by the averaging kernel matrix. The mathematical expression of the retrieved profile x is given as follows.

*S10/ Pg 9, L21. Section 3.3 should end and new section 3.4 "Time series" (or something similarly named) should start. The content from L22 onwards (to end of the section) is concerned with the time series, not explicitly error estimation. The sentence starting "Concentrations of CH4..." (Pg 6, L1) should be moved into this new section.*

Response on S10/pg 9, L21: The following subsection has been added and it starts with a statement from pg 6, L1

"3.4. Time series, partial column."

"Concentrations of $CH_4$ and $N_2O$ were derived from 166 spectra of NDACC filter 3 recorded from Dec. 2009 to March, 2013."....

*S11/ Figure 6 and Pg 6, L1. Please state the reason why is only data from 2009 to 2013 is analysed? I assume the Addis Ababa station is still currently (up to 2019) taking measurements?*

Response on S11/Figure 6 and page 6, L1: It has not worked since 2015 and we have taken the measurements for that period of time.

*S12/ Section 4: this section details MIPAS, MLS and AIRS satellite-based measurement platforms. It would be more helpful if the focus of this section was on details about Addis Ababa overpasses for each platform (such as the number of 'good' overpasses as a proportion of total). This would also help diagnose if Addis Ababa is a 'good' site (as the authors have stated) for such satellite validation.*

Response on S12/ Section 4:  The time when the satellites cross Addis Ababa has to be stated under each subsection. This May use to decide the importance of the site.

*S13/ Pg 11, L1. Why was only the period Mar2009 to Dec2010 used in MIPAS Addis Ababa comparisons? Why not longer?*

Response on S13/pg 11, L1: This work has prepared before three years that is why we used those limited time periods. The following statement has also been added on P10, L8 to explain the time period used to validate FTIR with MLS.

The comparison of FTIR with MLS for a time period of May 2009 to February 2013 has also made.  Here, in this paper we did not put anything about spatial-temporal variability of both $CH_4$ and $N_2O$

*S14/ Pg 12, L3. The last two sentences, starting with "Nitrous oxide derived…" should be omitted as it refers to MLS data version 2.2, not 3.3, unless the authors state (after verifying) that the precision of MLS N2O v3.3 is the same as v2.2.*

Response on S14/pg 12, L3: The last two sentences have been written as follows:

MLS N2O v2.2 has been validated and its precision and accuracy is respectively in Lambert et.al. (2007). The authors reported that MLS N2O precision is 24-14 ppbv (9-41%) and the accuracy is 70-3 ppbv (9-25%) in the pressure range 100-4.6 hPa.

*S15/ Section 5 details and quantifies comparisons between the satellite data products and the Addis Ababa ground based FTIR data, but (in my opinion) does not elaborate on the results with respect to other ground based FTIR site measurements. Are the biases and spread seen at Addis Ababa like that of other ground-based FTIR sites (most likely also part of the NDACC)? This would help ascertain if the Addis Ababa is a 'good' validation site and is network comparable. All that is required is a literature review, this will help put the results derived in this study in context.*

Response on S15/ Section 5: We were set the statement due to the results of the validation and to make it clearer, it has been replaced by  "Therefore, the performance of the FTIR instrument in capturing CH4 and N2O values at the Addis Ababa station is vital to monitor and understand the atmosphere over Addis Ababa. In addition the ground based observation has been used to supplement the satellite observations."

*S16/ Equations 4 to 10 all pertain to statistical calculations between the FTIR and MIPAS measurements. I assume the same statistical methods are applied to comparisons with the other satellite data? Maybe make this section more generic, not just MIPAS specific.*

Response on S16/ Equation 4 to 10: The paragraph and equations have been rewritten as follows; $sat_i$ (z) has been changed to $X_s$ (z) in equation 4.

The ground based FTIR measurements of CH4 and N2O has been validated at different locations (e.g. Senten et al. 2008). MIPAS, MLS and AIRS have a better vertical resolution than ground-based FTIR profiles and high temporal and spatial coverage in the tropics. The analysis of the comparison between volume mixing ratio values derived from FTIR and MIPAS were performed for the data sets collected on March 2009 to December 2010. Furthermore, the comparison of FTIR (CH4, N2O) with a MLS (CH4, N2O) and AIRS (CH4) for the time period of May 2009 to February 2013 has also made. Hence, the profiles from MIPAS, MLS and AIRS have been degraded to make a comparison between the FTIR and satellite observations. Therefore, the satellite measurement profiles are smoothed using the FTIR is averaging kernels of individual species obtained from the ground based FTIR retrieval by applying the procedures reported in Rodgers and Connor (2003 and given as

$$X_s = X_a + A(X_i + X_a)$$

The absolute difference at each altitude layers of a pair profile is calculated using

$$\delta_i(z) = [FTIR_i\ (z) - X_s(z)]$$

Where $FTIR_i\ (z)\ and\ X_s(z)$ are the FTIR and smoothed satellite profiles of CH$_4$ or N$_2$O respectively. The mean squares error can be expressed as

$$MSE_i = \sqrt{\frac{1}{N(z) - 1} \sum_{i=1}^{N(z)} \delta_i(z)^2}$$

$$\Delta_{rel}(z) = \frac{1}{N(z)} \sum_{i=1}^{N(z)} \delta_i(z)$$

Where $\delta_i$ (z) is the difference (absolute or relative), N(z) is the number of coincidences at z, $FTIR_i$ (z) is the FTIR VMR at z and the corresponding $X_s$(z) smoothed volume mixing ratio derived from satellite instruments.

*S17/ Pg 14, L3. "Hence we will focus on the random uncertainties associated with…". This statement does not connect with the analysis in section 5.2. In section 5.2 dataset biases are quantified, which includes both random and systematic uncertainties (not separated). The standard deviations of the dataset comparisons will also include any systematic uncertainties. Maybe this sentence be retracted or changed to explain what is meant in a clearer manner.*

Response on S17/pg 14, L3: To make this clear, we rewrite as follows and took only the random uncertainty of the instruments to evaluate the uncertainty of the comparison.

Here in this paper coincidence and smoothing errors are not taken into account in the full error analysis of the comparisons between remotely sensed data sets (von Clarmann, 2006). Hence, we focus on the random uncertainties of each instrument (Combined random error) that has been used to evaluate the uncertainty of the comparison (standard deviation of the difference).

S18/ Pg 14, L4. "However, the residual coincidence and horizontal smoothing errors...". If they are important why are they not investigated? The sentence starting on L19, pg14 ("In addition, the overestimation") also alludes (and offers conjecture) to issues arising around differences in the datasets relating to coincidence criteria but is not quantified. The authors could easily check this by changing the coincidence criteria (spatial and temporal) and see the effect of this in the dataset statistical differences.

Response on S18/pg14, L4: As the referee has suggested, the residual coincidence can be shown and we try to include on the reviesed manuscript.

S19/ Section 5.2 and  Section 5.3. In both these sections there is no mention of how dataset degrees of freedom affect the profile differences. For example, the ground-based retrievals of CH4 have approx. 2 DOFs. Differences at different altitudes will not be independent pieces of information. At Pg 14, L25 the authors state the bias of FTIR and MLS CH4 at 18-20km is insignificant, at 17km -1.7%, and between 20-27km below 11%. Are these pieces of information independent? The authors may wish to comment on this fact and its implications.

Response on S19/ Section 5.2 and Section 5.3: No, this information is dependent

S20/ Figures 10 and 11. There is no commentary on the large 'RD' differences above 30km (no sensitivity?). Could the authors comment on this?

Response on S20/Figures 10 and 11: As the sensitivity of the FTIR measurement is below 27 km that indicates the retrieved value above 27 km might be from the a priori. This makes the RD large above 30km.

S21/ Pg 17, L1. There needs to be a new section "section 5.4: Comparisons of partial columns" (or similar) starting at pg17 L1 if the authors are to  start discussing partial column comparisons. Currently, the partial column comparisons for both N2O and CH4are under section 5.3.

Response on S21/pg 17, L1: We have added the section 5.4 as suggested by the referee.

S22/ Pg 17, L1. Why are only MIPAS partial column comparisons conducted? What about other satellite data products?

Response on S22/page 17, L1:We have only taken the MIPAS partial column comparison due to time limitation. However, this may consider during submitting the revised manuscript.

*S23/ Fig 12. Uncertainty/error bars could be added to all data points. This would help in assessing the comparisons.*

Response on S23/Fig 12: The middle panel of all the comparison figures have a standard error of the difference (SEMAD, blue dotted).

*S24/ Pg 19, L12. Define 'good', do the authors mean the measurement quality and retrievals are 'good', or the location, or both? Since the focus of the manuscript is on assessing the performance of the Addis Ababa FTIR measurements, explaining 'good' is quite important.*

Response on S24/page 19, L12: The sentence on page 19, L1, "Therefore, the performance of instruments, FTIR, MIPAS and MLS in capturing CH4 and N2O values at the Addis Ababa station is good to study tropical atmospheric constituents." has been rewritten as follows. Since, the paper concerns on the FTIR measurement.

Therefore, the performance of the ground based FTIR instruments in Addis Ababa, Ethiopia site would be vital to fill the scarcity of ground based FTIR measurements at tropics as the bias obtained are in agreement to other sites.

***Technical comments (no particular order):***

*T1/ Title: FTIR should be expanded, not an acronym. There is no need for the chemical formulas. The word 'measurements' should also be added after FTIR. So…I recommend the full title should be along the lines of: "Methane and nitrous oxide fromground-based Fourier transform infrared spectrometer measurements at Addis Ababa: observations, error analysis and comparison with satellite data."*

Response T1, title: The title has been changed to Methane and nitrous oxide from ground-based Fourier Transform Infrared spectroscopy measurements at Addis Ababa: observations, error analysis and comparison with satellite data.

*T2/ Pg 1, L2. Possible change: "total column abundances and vertical distribution of various constituents in the atmosphere" to "total column trace gas abundances and vertical distributions".*

Response on T2, P1, and L2: have been replaced by "total column trace gas abundances and vertical distribution."

*T3/ Pg 1, L4. The superlative sentence "They reveal the high quality of FTIR measurements at Addis Ababa" is not required. The data and analysis reveal this.*

Response on T3, P1, L4: has been changed to "The data and analysis reveal the high quality of the FTIR measurements."

*T4/ In the abstract, I do not think it is necessary to specify satellite data product versions, for example 'V5R_CH4_224'. This is done in the main body.*

Response on T4: The importance of describing the version of the data on the abstract only to make it clear the readers with what version of the data has been validated. However, if the reviewer does not agree, we change it on the revised manuscript.

*T5/ Pg 1, L12. There are phases throughout the manuscript of the sort "a positive bias of less than 0.14 ppmv (9%) is found in the altitude range of 21to 27 km". I gather this means there is a maximum positive bias of 0.14 ppmv in the range 21 to 27km? This may be a better way to state it.*

*T6/ Pg 2, L1. CH4, N2O and CFCs are also stratospheric species…;)*

Response on T6, P2, L1: has been rewritten as follows

"Methane (CH4), nitrous oxide (N2O) and chlorofluorocarbons (CFCs) are the main source gases to the chemical families $NO_x$, $ClO_x$, and $HO_x$ at troposphere and stratosphere (Jacobson, 2005)."

*T7/ Pg 2, L7. "ENIVSAT" to "ENVISAT satellite".*

Response T7, P2, L7: has been added a word following "ENVISAT" "satellite"

*T8/ Pg 2, L9. Remove the word 'recent' from "The recent increasing…" (also replace 'to the' with 'on').*

Response T8, P2, L9:Done

*T9/ Pg 2, L10. Merge the sentences to read: "The recent increasing impact of CH4and N2O to the global warming has also been assessed by the last AR4 IPCC report (IPCC, 2007; Sussmann et al., 2012), additionally N2O will become the dominant ozone depleting substance emitted in the 21st century (Ravishankara et al., 2009)."*

Response T7, P2, L9-11: the sentence has been changed to

The increasing impact of CH4 and N2O on global warming has also been assessed by the last AR4 IPCC report (IPCC, 2007; Sussmann et.al. 2012), Additionally $N_2O$ will become the dominant ozone depleting substance emitted in the 21st century (Ravishankara et.al. 2009).

*T10/ Pg 2, L11. What is IASI? Expand to say: "IASI instrument aboard the MetOp-2 satellite". MetOp-1 or MetOp-2…I can't remember.*

Response T10, P2, L11: The sentence has been started with the following

"The Infrared Atmospheric Sounding Interferometer (IASI) on-board METOP-1 ……"

*T11/ Pg 2, L18. Rephrase first sentence: "In the tropics two important…"*

Response T11, P2, L18: The sentence has been rephrased as follows.

Two important exchange processes are taking place in the tropics, those are the inter hemispheric exchange and entry of tropospheric air mass to the stratosphere (Petersen et.al. 2010; Fueglistaler et.al. 2009).

*T12/ Pg 2, L25. Replace 'launched' with 'taken'.*

Response to T12, P2, L25: "launched" has been replaced by "taken"

*T13/ Pg 2, L27. Replace "The quality of ground based FTIR measurements" with "The Addis Ababa FTIR measurements".*

Response on T13, P2, L27: has rewritten as follows

The Addis Ababa FTIR measurement of atmospheric trace gases and their use to understand various lower and middle atmospheric processes have been reported in a number of previous studies (Takele Kenea et.al., 2013; Mengistu Tsidu et.al., 2015; Schneider et.al., 2015, 2016; Barthlott et.al.,2017).

*T14/ Pg 2, L33. Replace 'confirm' with 'show'.*

Response in T14, P2, and L33: "confirmed" has been replaced by "showed"

*T15/ Pg 2, L34. Replace 'biased high and provided +14% as the most likely bias' with 'biased 14% high'.*

Response on T15, P2, L34: The sentence has rewritten as follows

Laeng et.al. (2015) found the MIPAS CH4 profiles V5R_CH4_222 below 20 to 25 km biased +14% high

*T16/ Pg 2, L33. The reference Kenea throughout the manuscript should be replaced with Takele Kenea (2013)?*

Response: "kenea" has been replaced by "takele kenea" though out.

*T17/ Pg 2, L33. The quoted references of Laeng (2015) and Plieninger (2016) refer to MIPAS comparisons with other satellite products. The paragraph starting at Pg 2, L25 concerns Addis Ababa FTIR measurements. There is a jump in topic. Reading as is, it could easily be taken that Addis Ababa measurements were used in these studies. These sentences should be removed or moved to a different part of the manuscript.*

Response on T17, P2-3: The last two statements on paragraph 4 of introduction have moved to the subsection 4.1, since those sentences have been briefed about the bias of MIPAS measurements.

*T18/ Pg 3, L3. The sentence "In this study, the previous work on intercomparison is extended to source gases CH4and N2O from ground-based FTIR" is quite ambiguous. Either remove or make more specific to Addis Ababa.*

Response on T18, P3, L3: To make it clear, the statement has been rephrased as follows

In this study, the previous work on intercomparison of ozone (Takele Kenea et al., 2013) and water vopour. ((Samuel Takele Kenea., 2014)) are extended to source gases CH4 and N2O from ground-based FTIR of Addis Ababa site.

*T19/ Pg 3, L7. "approach" can be replaced with 'strategy'.*

Response on T19, P3. L7: The word "approach" has changed to "strategy"

*T20/ Pg 3, L13. Is Addis Ababa part of the Network for the Detection of Atmospheric Composition Change (NDACC)? I suspect so, if this is the case it should be stated. The Takele-Kenea paper should be used as a site reference paper.*

Response on T20, P3, L13: Our FTIR observatory has not registered yet as part of NDACC. Moreover, the reference has been added as suggested by the referee.

*T21/ Pg 3, L15. Could the Addis Ababa site altitude (MASL) also be added?*

Response on T21, P3, L13: Done

*T22/ Pg 3, L15. How is 'suitability' defined? Why is it suitable? Could possibly mention the amount of cloud free days a year.*

Response on T22, P3, L15: It is known that Ethiopia is over 250 days per year is cloud free.

*T23/ Pg 3, L18. The superlative "extremely" can be removed, not needed.*

Response on T23, P3, L18: Done

*T24/ Pg 3, L23. The superlative "very successfully"can be removed, not needed.*

Response on T24, P3, L23: Done, we have been rewritten the sentence as

"Fourier transform spectroscopy has been applied successfully to study trace gases in the atmosphere by examining atmospheric absorption lines in the infrared spectrum from solar."

*T25/ Pg 3, L24. Replace 'sun' with 'solar'*

Response on T25, P3, L24: Done (see the above response)

*T26/ Pg 3, L27. The sentence "This technique…" should be moved to precede the sentence "The high resolution…"*

Response on T26, P3, L27: We exchanged the two sentences

This technique uses the Sun as a light source to quantify molecular absorptions in the atmosphere and then retrieve trace gases abundance. The high-resolution FTIR Spectrometer, Bruker IFS120M upgraded with 125M electronics, from the Bruker Optics Company in Germany was installed in May, 2009 at the Addis Ababa site.

*T27/ Pg 3, L28. "Using seven narrow band filters". Assuming Addis Ababa is an NDACC site, do the seven filters meet NDACC specifications?*

Response on T27, P3, L28: Yes

*T28/ Pg 3, L29. It is mentioned an InSb detector is used to take measurements in over the range: 1500-4400cm-1*

*. There is no mention of detectors used to measure down to 750cm$^{-1}$, as mentioned in the prior sentence. Are measurements taken below 1500cm$^{-1}$?*

Response on T28, P3, L29: We have removed "The spectral coverage of the IFS120M instrument at the Addis Ababa site is 750 - 4000 cm$^{-1}$ using seven filters." sentence from the manuscript.

*T29/ Pg 3, L31. Replace "we used PROFFIT V…algorithm", with "we used the retrieval code PROFFIT (Ver95)".*

Response on T29, P3, L31: Done

*T30/ Pg 4, L2. As the sentence reads, PROFFIT was developed to only retrieve CH$_4$ and N$_2$O. Could this sentence be corrected to reflect the fact PROFFIT was developed to retrieve multiple species.*

Response on T30, P4, L2: The phrases in green are added in the sentences below

It has been developed based on semi-empirical implementation of the Optimal estimation Method (Rodgers, 2000) to derive the VMR profiles and column amounts of multiple species. Hence, CH$_4$ and N$_2$O profiles from measured spectra in the microwindows that span spectral range of 2400 - 2800 cm$^{-1}$ have been discussed in this paper.

*T31/ Pg 4, L3. The sentence "This algorithm…" only tells half the information. Once a forward model calculation is completed, what happens next?*

Response on T31, P4, L3: "to produce the synthesized spectra" is added before we closed the statement.

*T32/ Pg 4, L4. At the end of the sentence "The vertical profiles…N2O respectively" change to "N2O respectively (see table 1 for spectral regions)." and could possibly be moved to section 3.1.*

Response on T32, P4, L4: This has done *by taking* "The vertical profiles over Addis Ababa have been obtained by fitting five and four selected spectral regions for CH$_4$ and N$_2$O respectively (see table 1 for spectral region)." to P5 L6.

*T33/, Pg4, L6. Could the bottom (base) layer height be stated.*

Response on T33, P4, L6: We have added ", 2.45 to 85 km" after 41 layers to show the lower and top level of the layers.

*T34/ Pg4, L27. replace 'setup' with 'strategy'*

Response on T34, P4, L27: Done

*T35/ Pg 4, L29. This is the first time the 'NDACC' and 'IRWG' acronyms are used, please state in full.*

Response on T35, P4, L29: We agree that NDACC that has been stated in page 5 would be changed to page 4, L29.

*T36/ Pg 5, L4. I think the sentence "The vertical variability..." is not required, or more informationis required, i.e. define 'large'.*

Response on T36, P5, L4: As the referee suggested to delete Figure 1 and to make the explanation more clear, the paragraph has been replaced by "Both methane (CH4) and nitrous oxide (N2O) are well-mixed in the troposphere and their VMR decrease with height and becomes negligible with no variation above 55 km. The vertical variability of N2O and CH4 in the lower stratosphere is characterized by somewhat higher vertical gradient as compared to the other layers."

*T37/ Pg 5, L8. "The micro-windows have been adopted from different sources". Why, and could the sources, please be referenced.*

Response on T37, P5, L8: We mean, micro windows that give us a minimum residuals and errors were considered. The references from where the micro windows are adopted has been added at P5, L8.

The microwindows have been adopted from different sources (.Senten et al. 2008; Sussmann et al., 2011; Arndt et al., 2004).

*T38/ Pg, 5, L9. Modified Microwindows: why is this?*

Response on T38, P5, L9: The reasons why we modify the micro windows are due to high residuals obtained between the measured and synthesized spectra at the Addis Ababa site. Thus, the micro-windows recommended by the NDACC might be useful for the other FTIR sites found in mid and high latitudes. The references from where the micro windows are adopted has been added at P5, L8.

The microwindows have been adopted from different sources (.Senten et al. 2008; Sussmann et al., 2011; Arndt et al., 2004).

*T39/ Pg 5, L7. After the end of the sentence "lines are presented", the sentence Pg 4, L4 "The vertical profiles..." should be inserted.*

Response on T39, P5, L7: The sentence has added

The vertical profiles over Addis Ababa have been obtained by fitting five and four selected spectral regions (microwindows) for $CH_4$ and $N_2O$ respectively.

*T40/ Pg 5, L12. Remove the word "example". Also, could the authors detail the Signal to Noise ratio (SNR) of the Addis Ababa spectra. All this information is part of the 'optimization process'.*

Response on T40, P5, L12: Done

T41/ Table 1. could "int. Gases" be replaced with "Interfering gases". In the table legend, could "column amounts" be replaced with "total column amounts". Just an idea, (authors discretion), could another column be added for DOFs of the 0-27km partial column.

Response on T41, Table 1: Done

T42/ Figure 2. The usual convention is for measurement spectra data to be displayed as points (usually joined) and simulations as thin lines. The authors have the opposite of this. A minor point, and up to the authors if they would like to change it a more standard convention.

Response on T42, Figure 2: Done

T43/ Figure 2 Legend. Acronyms are not explained prior, so the full name is required. Sorry. SEA is an unusual way to label/present the solar angle. Does SEA mean Solar elevation angle? The standard convention is solar zenith angle (SZA = 90.0 - SEA). I recommend that SZA be presented, not SEA. Spectra time is presented as "101715", please reformat to 10h17m15s or similar. Is SD the root mean square difference of the measured spectra and forward model? I.e. RMS. Please define.

Response on T43, Figure 2:  Yes, the solar zenith angle is $20.6^O$ but we expressed in terms of solar elevation angle $69.4^O$. All the referee comments has taken to the caption of figuer 2 and 3. The RSM is defined as a set of values is the square root of the arithmetic mean of the squares of the values,

T44/ Pg 7, L2. To be pedantic, "the most" can be replaced with "an" (as a matter of opinion).

Response on T44, P7, L2: Done

T45/ Pg 7, L11. The authors should state at this point the units of the AVKs displayed, normalised to layer VMR or not? i.e. [VMR/VMR]

Response on T45, P7, L11: Figure 4. Has been changed to the selected altitude of AVKs and its unite has shown in the figure (x-axis).

T46/ Pg 7, L16. Pedantic point, but any AVK that has non-zero elements has 'sensitivity', that is, infer information from the spectra. 0.5 is an arbitrarily defined 'cut-off'. So, I think a better statement would be "Fig 4 (top panel) shows that the.... has  a sensitivity greater than 0.5 over the altitude range 2.45km to 27km".

Response on T46, P7, L16:  Changed

Fig 4 shows a strong sensitivity in the altitude range of the troposphere and lower stratosphere , i.e. 2.45 up to 27 km for the retrieval of $CH_4$ and $N_2O$.

T47/ Pg 8, L2. Could the altitude ranges of the two independent partial columns be stated.

Response on T47, P8, L2: It is the troposphere, below 17 km and the altitude range 18-27 km.

*T48/ Pg 8, L4. The sentence "The amplitude..." repeats information given earlier so can be removed.*

Response on T48, P8, L4: It was not explained in the FTIR and retrieval section.

*T49/ Pg 8, L5. The sentence "We also ignore": SorryI cannot understand this sentence (includes spelling mistakes), can it please be reworded and made clearer. The cited reference (Rinsland, 2005) does not contain any such information pertaining to altitude resolution. Can the authors please check the reference is correct or point out where it is in the paper.*

Response on T49, P8, L4:

*T50/ Figure 4 legend. The date time stamp of spectra the analysis is performed on should be given.*

Response on T50, Figure 4: The figure has changed  and the date has also shown in the caption. (see revised manuscript on page 8).

*T51/ Pg 9, L4. Could the acronyms for sources of error be included, i.e. 'instrument line shape (ILS)', so that they correspond to the legend labels in fig5.*

Response on T51, P9, L4: Yes, it was included in the figure and represented by a red line, but it was overlapped by other lines.

*T52/ Pg 9, L5. Can the term 'zero baselines offset' be 'zero level baseline offset'*

Response on T52, P9, L5: yes, it is zero level baseline offset.

*T53/ Pg 9, L5. I assume statistical error means 'random' error, maybe rephrase as "statistical (random) error". Also take out "typical" as it relates directly to an example.*

Response on T53, P9, L5: Done

*T54/ Pg 9, L17. Change part of the sentence "for the profile", to "for the N2O profile".*

Response on T54, P9, L17: Done

*T55/ Pg 10, L6. The sentence starting "Vertical resolution…" needs to be referenced.*

Response on T55, P10, L6:  The vertical resolutions that has been stated in the discussion manuscript was for the old version. This has replaced by

The vertical resolution of MIPAS ranges from 2.5 to 7 km for CH4, and from 2.5 to 6 km for N2O in the reduced-resolution period (Plieninger et al., 2015).

*T56/ Pg 10, L8. The sentence starting "The analysis of the comparison…" should be moved to section 5.1.*

Response on T56, P10, L8: As the referee has stated, the expression has already in section 5.1 and then we have removed it from this subsection. (see the response on S16).

*T57/ Pg 11, L17. The starting sentence should be reworded: "...we have used the MLS N2O (v3.3) product to validate the ground-based Addis Ababa FTIR measurements."*

Response on T57, P11, L17: The sentence in pg 11, L17 has been replaced by

"MLS N2O data set has been used to validate the ground-based FTIR measurements. However, methane ($CH_4$) data are derived using coincident measurements of atmospheric water vapor ($H_2O$), carbon monoxide (CO) and nitrous oxide (N2O) from the EOS MLS instrument on the NASA Aura satellite and detail are given in Minschwaner et al. (2015)."

*T58/ Pg 11, L18. The altitude levels are given in pressure coordinates. So far in the manuscript, the altitude units have been in kilometres. Could the authors include the geometric altitude as well as the pressure. I.e. "100 and 0.1 hPa (XX to XX km's)"*

Response on T58, P11, L18: those altitude ranges are showing only for the data set and in the comparison we put them in km as it is known that the altitude can be expressed both in pressure and km, such as 100-0.1hpa mean 17-63 km.

*T59/ Pg 12, L7. This sentence needs a reference.*

Response on T59, P12, L7: The sentence has rewritten as follows and reference was also added.

Operating in nadir sounding geometry, the Atmospheric Infrared Sounder (AIRS) on board the Aqua satellite launched into Earth orbit in May 2002 Chahine et al. (2006).

*T60/ Pg 12, L11. Unfortunately, the sentence starting "The spectral resolution…" does not make sense, could this be reworded, rephrased. The following sentence can be omitted as it does not add any information concerning this study.*

Response on T60, P12, L11: we changed the paragraph as follows.

Operating in nadir sounding geometry, the Atmospheric Infrared Sounder (AIRS) on board the Aqua satellite launched into Earth orbit in May 2002 Chahine et al. (2006). AIRS is a medium-resolution infrared grating spectra radiometer and a diffraction grating disperses the incoming infrared radiation into 17 linear detector arrays comprising 2378 spectral samples. The satellite crosses the equator at approximately 1:30 A.M. and 1:30 P.M. local time, resulting in near global coverage twice a day. AIRS 2378 channels covers from 649 to 1136, 1217–1613 and 2169–2674 $cm^{-1}$. It also measures trace gases such as O3, CO and to some extent CO2. AIRS $CH_4$ and $N_2O$ retrievals have been characterized and validated by Xiong et al. (2008) and Xiong et al.(2014) respectively.

*T61/ Pg 12, L13. Could the data product version of AIRS CH4 be added?*

Response on T61, P12, L13: AIRS version 6

*T62/ Pg 12, L20. The word 'version' can be removed.*

Response on T62, P12, L20: Done

*T63/ Pg 12, L22. I think the meaning of 'degraded' means smoothed, so maybe the sentence could read "…MLS have been degraded (smoothed) to make a …" or replaced degraded with smoothed.*

Response on T63, P12, L22: Done

*T64/ Pg 12, L29. Replace 'parameters' with 'statistics'.*

Response on T64, P12, L29: Done

*T65/ Pg 13, L11. I assume that Sati(z) is smoothed? So maybe state that: "and the corresponding Sati(z) smoothed volume mixing ratio is derived from…"*

Response on T65, P13, L11: Equation 4 to 10: The paragraph and equations have been rewritten as follows; $sat_i(z)$ has been changed to $X_s(z)$ in equation 4.(see Response on S16).

*T66/ Figures 7, 8 and 9 could be combined into a single 3x3 figure, so could figures 10 and 11 (2x2)*

Response on T66, Figure 7: They would not see clearly.

*T67/ Pg 14, L26. "In the tropopause layer", could the tropopause layer be defined, i.e. "In the tropopause layer (~XX-XX km)"*

Response on T67, P14, L26: It has been defined as 16-18 km. In addition, it varies in height according to season. Here, in this work we have already put it in the next expression 17 km.

*T68/ Figures 7,8,9,10 & 11. The legends in the figures are slightly different. Maybe standardise these as the figure captions all reference back to fig 7 caption. Figure 7 caption should also explain/define the legend captions. For example: "mean difference FTIR minus MIPAS (MAD, blue solid line) …"*

Response on T68/ Figures 7, 8 9 10 & 11: The legends in the figures have been corrected and all the legends have also stated in the caption.

Figure 7. Comparison of CH4 from MIPAS reduced resolution (V5R_CH4_224) and FTIR. Left panel: mean profiles of MIPAS (red) and FTIR (black) and their standard deviation (horizontal bars). Middle panel: mean difference FTIR minus MIPAS (MAD, blue solid), standard error of the difference (SEMAD, blue dotted), and mean relative differences FTIR minus MIPAS relative to their averaged (MRD, green, upper axis). Right panel: combined mean estimated statistical error of the difference (combined error, red dotted, contains MIPAS instrument noise error and FTIR random error budget), standard deviation of the difference (STDMAD, black solid).

*T69/ Pg 16, L2. The first sentence could be abbreviated to "FTIR N2O mixing ratio MIPAS comparison results are shown in fig 10."*

Response on T69, P16, L2: Done

*T70/ Pg 16, L8. (-0.02 ppmv) ... include units please.*

Response on T70, P16, L8: Done

*T71/ Pg 16, L11. Replace 'can' with 'could'.*

Response on T71, P16, L11: Done

*T72/ Pg 16, L17. Add to end of sentence: "the value derived from the FTIR is overestimated (relative to MLS)"*

Response on T72, P16, L17: Done

*T73/ Pg 18, L4. The last part of the first sentence "…, which is a very useful…" is not needed.*

Response on T73, P18, L4: has been corrected

*T74/ Figure 12. The date label lacks information on the years(s), only months are given. Add information on the year(s) comparisons were made.*

Response on T74, Figure 12: It has been corrected by adding May 2009 to December 2010.

*T74/ Acknowledgement: Remove full stop at the start of the first sentence. Remove the word 'besides'. Support, not supports.*

Response on T74: Done

*T75/ References. The authors may wish to take out, or update references concerning an ACPD articles, such articles have not passed the peer review process. Reference formatting differs, so would be good to get it all consistent. Decide on an author convention for Samuel Takele Kenea, as this author is referenced a few times, but referenced differently (Takele Kenea S.).*

Response on T75, References: All the reference with ACPD, Atmos. Meas. Tech. Discuss. and in naming Takele Kenea has been corrected.

---

## Referee Report (RR1)

**Comments on Authors reply/changes to the manuscript: amt-2019-170 for the Editor.**

Hi there,

I have received and read the authors reply/rebuttals (amt-2019-170-AC1-supplement.pdf and amt-2019-170-AC2-supplement.pdf) to reviewer's comments on the manuscript amt-2019-170.pdf.

Unfortunately (and not taken lightly) I cannot recommend the current altered manuscript to be published. There are still too many mistakes. Reviewers comments were not adequately addressed.

Concerns:

-Many of the replies to both reviewer's comments and stated manuscript changes were incomplete with spelling mistakes (not a good start). For example: amt-2019-170-AC1-supplement.pdf P5, L1 reply: incomplete.

*P5, L1: Why different versions of HITRAN were used. Please explain and also, what versions were used for gases*
Response on P5, L1: As the referee suggested, we added sentences that description which versions of HITRAN data were used for gases. Different versions of HITRAN were used as the new updated HITRAN……………….
Required

-The authors were inconsistent in illustrating manuscript changes. Sometimes the authors stated the changes verbatim, along with page and line reference (as per protocol) but other times the reply only stated changes were made, not the actual change and location. This makes it hard to conceptualise all the changes to the manuscript or find out the actual changes.

*S5/ Pg4, L28: Apriori is mentioned. Are the apriori profiles used static? i.e. unvarying, or are they changing seasonally, yearly, or daily? If the apriori is static, then how is it constructed, a mean over XX years? Is the apriori based on a certain global region?*
Response on S5/pg 4, L28: It was static, mean of 40 years WACCM for tropics was used.

-Reasons were not given to some questions. Example below:

*P5, L15. It is mentioned that the retrieval strategy is optimized using a single spectra: : : please expand this description, what do you use a criteria for optimization? Is it consistent for all months, zenith angles?.*
Response on P5, L15: The optimization of the retrieval strategy starts from the selection of the micro windows which are somewhat different from the recommended by NDACC.

-There was a lack of adequate response to very important questions. Examples:

    -How the Tikhonov regularization scheme was formulated and the requirement for extra Tikhonov details:

    *S8/ Pg 5, L15. The authors mention an "optimised retrieval strategy" but only give a passing mention to the Tikhonov retrieval regularization scheme. This is an important part of the retrieval; influencing overall information content and interlayer correlations of information content. Could the author please describe the Tikhonov regularization parameters. Why was the Tikhonov scheme implemented instead of using apriori uncertainties? What type of smoothing constraint is used (L1, L2 etc..), were the smoothing constraints normalised using layer thickness? what is the alpha*

*parameter used? and how was the alpha parameter selected? is the alpha parameter static? or varies per retrieval?*

Response On s8/page 5, L15: detailed description of retrieval strategy has been added after L10.

Methane and nitrous oxide vertical profiles over Addis Ababa have been obtained by fitting five and four Micro windows respectively. The retrieved state vector contains the retrieved volume mixing ratios of the target gas defined in 41 layers of the tropical atmospheric conditions. The retrieved profiles were derived using a Tikhonov-Phillips method on a logarithmic scale.

-The reasons for different micro-window selection to that of standard NDACC IRWG practice was not given:

*T38/ Pg, 5, L9. Modified Microwindows: why is this?*

Response on T38, P5, L9: The reasons why we modify the micro windows are due to high residuals obtained between the measured and synthesized spectra at the Addis Ababa site. Thus, the micro-windows recommended by the NDACC might be useful for the other FTIR sites found in mid and high latitudes. The references from where the micro windows are adopted has been added at P5, L8.

-Addressing the concerns around profile comparisons when there is only ~2 DOFs:

*Section 5.2. It is not explained why authors compare FTIR vs satellite vertical profiles. The FTIR information content is limited to 2 DOFs (tropospheric and stratospheric columns) but main figures for the comparison are shown as profiles.*

Response, Section 5.2.: Since those results are the first for Addis Ababa.

-Investigating spatial-temporal co-location criteria on the measurement comparisons which could possibly affect comparison results:

*P10, L8. Explained why only satellite data between March 2009 to Dec 2010 is used for MIPAS?, what about the other satellite measurements (it is not mentioned)? why this wide range is used. I would try other distances as well. How do you assess the spatial-temporal variability of both $CH_4$ and $N_2O$.*

Response on P10, L8: We have added the statement to explain the period time used to validate FTIR with MLS.

The comparison of FTIR with MLS for a period time of May 2009 to February 2013 has also made. Here, in this paper, we did not put anything about the spatial-temporal variability of both $CH_4$ and $N_2O$.

-To a lesser extent, I also feel the authors have missed an opportunity to explain the aims of the study better, and the importance of the Addis Ababa site location and measurements made.

*The authors state that the comparisons at the "Addis Ababa station is good to study tropical atmospheric processes" (Pg 19, L12). „Good" in what context? Given the comparison results, will the ground-based $CH_4$ and $N_2O$ measurements capture seasonal cycles and multi-year trends? Will biomass burning or other episodic events most likely be seen, and from what part of the tropics (the tropic is a large place)?*

Response on S2: Since this result is the first to the Addis Ababa site, it required validation to assess the quality of the ground-based FTIR measurements. The motivation is to fill the gap in understanding the atmosphere over Addis Ababa, tropical region of the globe. Moreover,

there are scores of measurements in tropical. The expression below has been added in section 2.1 to elaborate more about the measurement site.

Ethiopian is characterized by high rainfall and temperature variability on both spatial and temporal scales. The variability in distribution is related to altitude, latitude, humidity and winds, which are the significant factors in affecting the weather system of the country.

I am still willing to review/comment on any future manuscripts. It is a shame as the content and overall aim and methodology of the manuscript is robust and would be a welcome addition to the literature.

---

## Referee Report (RR2)

**Comments on Authors reply/changes to the manuscript: amt-2019-170 for the Editor.**

Yirdaw berhe, et al., have resubmitted their manuscript comparing ground-based MIR-FTS measurements of atmospheric $CH_4$ and $N_2O$ at Addis Ababa, Ethiopia to that of three satellite (MIPAS, MLS and AIRS) data products.

I have received and read the authors reply/rebuttals (amt-2019-170-author_response-version2.pdf), along with the new supplement, amt-2019-170-supplement-version1.pdf. The authors supplied an updated/revised manuscript: amt-2019-170-manuscript-version5.pdf. The authors have made substantial changes to the manuscript to address reviewer's concerns.

Unfortunately, the revised manuscript presented at the end of the rebuttal document amt-2019-170-author_response-version2.pdf (which has highlighted changes, thanks, very handy) and the revised manuscript amt-2019-170-manuscript-version5.pdf (with no changes highlighted) are different. For example, section 2.2 is different between these two manuscript versions. Which is the correct version? This makes reviewing difficult. All commentary below is based upon the document: amt-2019-170-manuscript-version5.pdf

Unfortunately (and not taken lightly), again, I cannot recommend the current altered manuscript (amt-2019-170-manuscript-version5.pdf) to be published. There are still too many spelling, grammatical and reference citation. Yirdaw berhe, et al. have addressed many of the initial reviewer's comments but I still find some critical replies/rebuttals are inadequate. Such inadequacies refer mainly to technical details, and once these are addressed (along with the mistakes) I would recommend publication. I do not see a major revision is required.

**Spelling, grammatical and reference citation mistakes:**

Throughout the revised manuscript amt-2019-170-manuscript-version5.pdf and the reply to authors document (amt-2019-170-author_response-version2.pdf) there are numerous grammatical and spelling mistakes. The formatting of cited references still uses more than one type of style. I recommend the authors perform more intensive copy editing themselves or employ a copy-editing service.

**Technical concerns:**

1/ The figures in amt-2019-170-supplement-version1.pdf are of a low visual standard. Could the figures be replaced with those of a correct aspect-ratio?

2/ Retrieval strategy and quality:

The theory of the Tikhonov retrieval strategy is provided in the supplement and (starting) on pg. 5, L22 of the revised manuscript. The details supplied do not address the questions of the reviewers on this topic. It is stated at pg. 6, ln 1 in the revised manuscript "An optimised retrieval strategy for tropics has been established", but there are no details given on the method or results of the optimised strategy. For example, what is the value of the Alpha parameter and how was this value decided upon. What is the $S_e$ value used? Also, the retrieval strategy is also optimised for Addis Ababa, not the for tropics in general.

On pg. 6, ln 7 the manuscript states "The magnitude of the residuals indicates that measured spectra…. of both $CH_4$ and $N_2O$ was quality as they are less than 1". Apart from grammatical mistake, the authors apply a QC/QA filtering limit of '1' to the residuals to assess retrieval 'quality'. There are no units of this value, or a description of the meaning of the value. Could the authors please explain in more detail the methods assessing retrieval quality.

On pg. 4, ln 25 the statement "The quality of the measurements during the time period of May 2009-February 2011 has revealed by Takele Kenea et al. (2013)." is out of place in this section and has no context. This statement should be at the end of section 3.1 and the authors should also give the readers more insight as to what the actual criteria are used to assess quality.

3/ I think the authors response and manuscript correction to the reviewer's question below is inadequate.

My understanding is that measurements started in 2009 to present, is this correct? Why you use limited number of years? I highly encourage to include more years and if possible trend analysis. Do the trend make sense?

Response on P10, L8: The statement at P 10, L 8 has been deleted and taking the information to P
12, L16.
• The FTIR measurement was not functional from March, 2011 to November, 2012.
• I think the instrument is not functional starting 2015.
Response: We have added the following statement in Pg 12, L 16 to explain the period time considered during the intercomparison of FTIR CH4 and N2O with MIPAS, MLS and AIRS.
The quality of the FTIR CH4 and N2O for a period that covers May 2009 to March 2013 is assessed through comparison with data from MIPAS (May 2009 to December 2010), MLS (May 2009 to March 2013) and AIRS (May 2009 to March 2013) sensors on board satellites.

The authors need to explicitly state why certain periods were excluded, i.e. the instrument was not functional between March 2011 and November 2012. Also, why comparisons halted at March 2013. Was the instrument not working from 2013 onwards? Or alternatively, why the period May 2009 to March 2013 was the period selected for satellite comparisons.

4/ Coincidence criteria.

The authors state "The more stringent latitudinal criterion has proven to be a good choice for all comparisons, since latitudinal variations are, in general, more pronounced than longitudinal ones" (pg. 12, ln 9). Could the authors explain why the chosen limits are more stringent (and compared to what?) and why they are a good choice.

5/WACCM citable reference: Pg. 4, ln 31. The WACCM model needs a citable reference. http://www2.cesm.ucar.edu/working_groups/?ref=nav does not refer to WACCM.

6/ pg.12 ln 18: "Hence, the profiles from MIPAS, MLS and AIRS have been degraded to make a comparison between the FTIR and satellite observations."

I think degraded is the wrong word, ambiguous. Could the authors rewrite or explain the term 'degraded' in a better way.

7/ Partial column altitude range in section 5.4 and Fig 11. Caption

I infer the partial column range is 15-27km? is this correct? Could the authors please explicitly state the exact range in section 5.4 and fig 11 caption.

---

## Author Response (AR2)

I thank the referees for giving me a second chance to revise my manuscript.

The abstract and conclusion has been re-written in the revised manuscript. Supplement document is also attached.

Our responses are given point-by-point below in blue following each of the reviewers' comments, which are repeated in full black. Reproduced text of the revised manuscript is set in green

**Responses to referee 1:**

Anonymous Referee #1

This work presents retrievals of CH4 and N2O using a ground-based FTIR instrument at Addis Ababa, Ethiopia with aim to present observations, error analysis, and comparisons with satellite data. The lack of long-term remote sensing observations at Addis Ababa makes this work important. The technique and results of such measurements might be interesting and likely suitable for the journal. However, I still have major comments and do not recommend the current manuscript; revisions are warranted before publication. The quality of the paper needs to be improved before publication.

Major Comments

I have the following major comments:

(1) In this revised versions authors state that the goal is twofold: (1) present a retrieval strategy of CH4 and N2O, and (2) validate the FTIR observations using satellite observations.

Regarding 1, the retrieval strategy applied in this work is different than the NDACC/IRWG recommendations. The authors claim that they use different micro windows because high residuals are obtained if using NDACC recommendations. However, this is not mentioned and shown in the manuscript. It would be valuable for the IRWG/NDACC community, if this in fact is true. By checking the NDACC archive I can see other sites within the latitude range of Addis Ababa and they use the recommended settings. I highly encourage to include a thorough analysis comparing both retrieval strategies (even as supplemental information). The current manuscript just mentions that they use different micro-windows and interfering species but they do not justify. A thorough analysis might consist in showing retrieval fit examples using both methods and at least some months' worth of data (e.g., linear correlation).

Response: I have attached a document as a supplement that explains the retrieved results of IRWG/NDACC and the micro windows used in this work.

Regarding 2. In the abstract authors state "They reveal the high quality of FTIR measurements at Addis Ababa" and then they show biases in the FTIR with respect to three different satellite measurements.

Response: I have added a table at Pg 15, L 6

Table 2. Averaged statistical means (M) and standard deviations (STD) of the relative differences 100* [(FTIR+ MIPAS)/(FTIR+MIPAS)/2][%] defined in altitude range of 17-20 km and 21-27 km. The

numbers of coincidences (N) within ±2 degrees of latitude and ±10 degrees of longitude and time difference of ± 24hr are selected for intercomparison. This is for FTIR CH4 and N2O with the corresponding other instruments (stated in second column).

| Gases | Instruments | Altitude range (kKm) | RD(%) ± STD.RD(%) | period | N |
|-------|-------------|----------------------|-------------------|--------|---|
| CH4 | MIPAS | 17-20/21-27 | -4.9/4.2 | May 2009-Dec. 2010 | 29 |
| | MLS | 17-19/20-27 | -1.8/5.8 | May 2009-Feb 2013 | 77 |
| | AIRS | 17-20/21-27 | -2.8/5.3 | May 2009-Feb 2013 | 118 |

In all the comparison of FTIR CH4 with data from MIPAS, MLS and AIRS sensors on board satellites indicates a negative bias below 21 km and a positive bias above 21 km with similar bias of not higher than 5.8 % in the altitude range 21-27 km (see Table 2.). The volume mixing ratio derived from the satellite are higher in altitude lower than 21 km.

First, A justification of why these three satellites is missing but highly important. This is crucial if authors aim to validate ground-based FTIR observations using satellite, which normally is the other way around.

Response:  The three satellite data (MIPAS, MLS and, AIRS) used in the comparison as they have better vertical resolution than ground-based FTIR profiles due to observation geometry, spectral windows and measurement techniques. The following information has been inserted in Pg 12, L 19:

The satellite data (MIPAS, MLS and, AIRS) used in the following comparisons have considerably better vertical resolution than ground-based FTIR profiles due to observation geometry, spectral windows and measurement techniques.

The aim of this study is to derive column abundances and profiles of CH4 and N2O from solar absorption measurements taken by FTIR at the tropical high altitude site of Addis Ababa, Ethiopia for a period that covers May 2009 to March 2013. The intercomparison with data from MIPAS, MLS and AIRS sensors on board satellites have been made to assess the quality of the data derived from FTIR.

Second, the main findings of the paper are summarized in the abstract as follow:

[revised manuscript text omitted]

I found the above text very confusing and ambiguous. I suggest re-arrange and try to explain better the main findings. Are the findings the same when comparing with the three satellites? it's not clear to me.

Response: To make clear the findings, i re-arranged the intercomparison results summarized in the abstract. In addition to the re-arrangement of the information, the following information has been inserted in Pg 1, L 18 and summarized in Table 2.

In all the comparison of FTIR CH4 with data from MIPAS, MLS and AIRS sensors on board satellites indicates a negative bias below 21 km and a positive bias above 21 km with similar bias of not higher than 5.8 %. Therefore, the bias obtained from the comparison and the precision of the FTIR measurements are comparable which allow the use of the data in further scientific studies as it represents a unique environment of tropical Africa, a region poorly investigated in the past

Furthermore, and probably most importantly, the authors compare and report biases at different altitude ranges from about 11 to 27km. I do not believe the ground-based observations are able to retrieve highly resolved vertical profiles. I expect one degree of freedom in the troposphere, where these satellites are not sensitive, and a second degree of freedom in the stratosphere. The abstract and the manuscript sounds like authors claim more degrees of freedom.

Response: We took only the intercomparisons where a negative and positive bias are found to show the altitude ranges where volume mixing ratio derived from FTIR is higher (positive bias) and lower (negative bias). In the cases of DOFs, we have already stated in Pg 12, table 1.

Lastly, authors introduce versions of satellite products using awkward names for a paper, e.g., IMK/IAA MIPAS_CH4_224/225, which I found very annoying/distracting.
Response: corrected as "MIPAS_CH4_224"
(2) My understanding is that measurements started in 2009 to present, is this correct? Why you use limited number of years? I highly encourage to include more years and if possible trend analysis. Do the trend make sense?

Response on P10, L8: The statement at P 10, L 8 has been deleted and taking the information to P 12, L16.

- The FTIR measurement was not functional from March, 2011 to November, 2012.

- I think the instrument is not functional starting 2015.

Response: We have added the following statement in Pg 12, L 16 to explain the period time considered during the intercomparison of FTIR CH4 and N2O with MIPAS, MLS and AIRS.

The quality of the FTIR CH4 and N2O for a period that covers May 2009 to March 2013 is assessed through comparison with data from MIPAS (May 2009 to December 2010), MLS (May 2009 to March 2013) and AIRS (May 2009 to March 2013) sensors on board satellites.

Specific Comments

Authors use HITRAN 2004, with 2009 and 2012 updates. Why HITRAN 16 is not used?, explain.

Response: This manuscript has been prepared before 3 years ago and even HITRAN 2004 was also not used in this work and rewritten as follows:

For the N2O retrievals, the spectroscopic line parameters from the HITRAN 2008 database including official updates through 2012 (Rothman et al., 2013). For the CH4 retrievals, HITRAN 2012 database was used (Rothman et al., 2013).

In the paper "coincident criteria of ±2◦ of latitude and ± 10◦ of longitude from the ground-based FTIR site in Addis Ababa and within time difference of ±24h". Please justify these criteria.

Response: The closest satellite measurements (on the same day as the ground-based FTIR measurements) within ±2 degrees of latitude and ±10 degrees of longitude are selected for intercomparison. The following information has been inserted in Pg 12, L 19.

The more stringent latitudinal criterion has proven to be a good choice for all comparisons, since latitudinal variations are, in general, more pronounced than longitudinal ones (Takele et.al., 2013). These criteria yielded 29, 77 and 118 days of coincident measurements between FTIR and MIPAS, MLS and AIRS respectively.

I highly suggest to review exhaustively the English along the manuscript.

Review links included in the manuscript, some do not work.

Responses to referee 2:

**Comments on Authors reply/changes to the manuscript: amt-2019-170 for the Editor.**

Hi there,

I have received and read the authors reply/rebuttals (amt-2019-170-AC1-supplement.pdf and amt-2019-170-AC2-supplement.pdf) to reviewer's comments on the manuscript amt-2019-170.pdf.

Unfortunately (and not taken lightly) I cannot recommend the current altered manuscript to be published. There are still too many mistakes. Reviewers comments were not adequately addressed.

Concerns:

-Many of the replies to both reviewer's comments and stated manuscript changes were incomplete with spelling mistakes (not a good start). For example: amt-2019-170-AC1-supplement.pdf P5, L1 reply: incomplete.

*P5, L1: Why different versions of HITRAN were used. Please explain and also, what versions were used for gases*
Response on P5, L1: As the referee suggested, we added sentences that describe which versions of HITRAN data were used for gases. This manuscript is taking time to prepared and at the beginning HITRAN 2004 were used, but finally we used HITRAN 2009 and the updated 2012.

The spectroscopic parameters were taken from the High Resolution Transmission (HITRAN) database version 2008 of N2O, 2009 for H2O (Rothmann et al., 2009) and the updated HITRAN 2012 for CO2, CH4, NO2 (Rothmann et al., 2013) were used during retrieval of CH4 and N2O.

-The authors were inconsistent in illustrating manuscript changes. Sometimes the authors stated the changes verbatim, along with page and line reference (as per protocol) but other times the reply only stated changes were made, not the actual change and location. This makes it hard to conceptualise all the changes to the manuscript or find out the actual changes.

Response: Sorry for not making my responses consistent, since I have gotten a difficulty on showing the

track change made on the manuscript. Here, I would show the changes made on the manuscript clearly using the page and line numbers of the original.

*S5/ Pg4, L28: Apriori is mentioned. Are the apriori profiles used static? i.e. unvarying, or are they changing seasonally, yearly, or daily? If the apriori is static, then how is it constructed, a mean over XX years? Is the apriori based on a certain global region?*

Response on S5/pg 4, L28: It was static, mean of 40 years WACCM for the Addis Ababa site was used. The following information has to be inserted in pg 4 L 29 in order to clearly explain the a priori applied in our retrieval strategy.

WACCM is a numerical model developed at the National Center for Atmospheric Research (NCAR). The a priori for the target gases and interference were constructed using the averaged values from the monthly WACCM profiles for a period that covers 1980-2020 that applied for Addis Ababa FTIR $CH_4$ and $N_2O$ retrievals.

-Reasons were not given to some questions. Example below:

*P5, L15. It is mentioned that the retrieval strategy is optimized using a single spectra: : : please expand this description, what do you use a criteria for optimization? Is it consistent for all months, zenith angles?.*
Response on P5, L15: All settings (micro-windows, constraint, initial guess and a priori profile) which are used for single spectra on this paper is considered it as example. Similarly, all the settings are applied on all the measured spectra.

-There was a lack of adequate response to very important questions. Examples:

-How the Tikhonov regularization scheme was formulated and the requirement for extra Tikhonov details:

*S8/ Pg 5, L15. The authors mention an "optimised retrieval strategy" but only give a passing mention to the Tikhonov retrieval regularization scheme. This is an important part of the retrieval; influencing overall information content and inter layer correlations of information content. Could the author please describe the Tikhonov regularization parameters. Why was the Tikhonov scheme implemented instead of using apriori uncertainties? What type of smoothing constraint is used (L1, L2 etc..), were the smoothing constraints normalised using layer thickness? what is the alpha parameter used? and how was the alpha parameter selected? is the alpha parameter static? or varies per retrieval?*

Response on Pg 5/L10: The information below has been inserted in pg 5/L10 of the first manuscript: (se also the supplementary document on retrieval strategy).

PROFFIT includes various retrieval options such as scaling of a priori profile, the Tikhonov-Phillips method (Phillips, ,1962; Tikhonov, , 1963), or the optimal estimation method (Rodgers , 2000). In this study, Tikhonov-Phillips regularization method on a logarithmic scale is used during the retrieval of CH4 and N2O. In case of Tikhonov regularization the matrix R which is the regularization or constraint matrix in the equation of iterative solution has been expressed by R= $\alpha L^T L$ , where $\alpha$ is a regularization parameter and L is a regularization matrix and the iterative solution have obtained an additional parameters($\alpha$, L). The retrieval is performed on a fine vertical grids from 2.45 to 85 km and is stabilized by a first order Tikhonov constraint, R= $\alpha L^T_1 L_1$, where is the strength of the constraint and L1 is the first order derivative (Borsdorff et al., 2014), which

smooths the solution without biasing it towards the a priori profile. The parameter  determines the weight of the regularization and it is also important to choose appropriate to the problem. One way to fix this parameter is the L-curve method (Hansen, , 1992). The regularization strength , is determined by finding a trade-off between the number of degrees of freedom (measure of amount of information in methane and nitrous oxide retrieval), which is given by the trace of the averaging kernel and the noise induced error (Rodgers , 2000). All settings (micro-windows, constraint, initial guess and a priori profile) are chosen in such a way that all the structures visible in the retrieved distributions originate from the measurements and are not artifacts due to any constraints. An optimized retrieval strategy for tropics has been established within the framework of this paper for the retrieval of CH4 and N2O by applying it first to single spectra as test cases, and later routinely to the full set of measurements.

-The reasons for different micro-window selection to that of standard NDACC IRWG practice  was not given:

*T38/ Pg, 5, L9. Modified Microwindows: why is this?*
Response on T38, P5, L9: The reasons why we modify the micro windows are due to high residuals obtained between the measured and synthesized spectra at the Addis Ababa site while we use the micro-windows recommended by the NDACC IRWG. Our tests using the selected micro windows are shown less residual (see the supplement material). The references from where the micro windows are adopted have been added at P5, L8.  The following information has been inserted in pg 5, L 10.

The main criterion for selection of thus microwindows is high sensitivity to methane and low interference from other gases. Our tests have shown that these windows are still appropriate for the Addis Ababa site.

-Addressing the concerns around profile comparisons when there is only ~2 DOFs:

*Section 5.2. It is not explained why authors compare FTIR vs satellite vertical  profiles. The FTIR information content is limited to 2 DOFs (tropospheric and  stratospheric columns) but main figures for the comparison are shown as profiles.*

Response, Section 5.2.: The comparison has been made to assess the quality of the data derived from solar absorption measurements taken at Addis Ababa, Ethiopia site.  Moreover, the retrieved profiles of CH4 and N2O are the first for the Addis Ababa FTIR observatory. In addition to the vertical profiles, partial columns of the two gases have also compared with MIPAS and discussed in the manuscript.  The DOFs for the altitude range in 15 – 27 Km is ~1 (see pg 9, L 29)

-Investigating spatial-temporal co-location criteria on the measurement comparisons which could possibly affect comparison results:

*P10, L8. Explained why only satellite data between March 2009 to Dec 2010 is used for MIPAS?, what about the other satellite measurements (it is not mentioned)? why  this wide range is used. I would try other distances as well. How do you assess the spatial-temporal variability of both CH4 and N2O?*

Response on P10, L8: The statement at P 10, L 8 has been deleted and taking the information to P 12, L16.

- The FTIR measurement was not functional from March, 2011 to November, 2012

Response: We have added the following statement in Pg 12, L 16 to explain the period time considered during the intercomparison of FTIR CH4 and N2O with MIPAS, MLS and AIRS.

The quality of the FTIR CH4 and N2O for a period that covers May 2009 to March 2013 is assessed through comparison with data from MIPAS (May 2009 to December 2010), MLS (May 2009 to March 2013) and AIRS (May 2009 to March 2013) sensors on board satellites.

Response: The following information has been inserted in Pg 12, L 19 so that to answere why we use that spatial-temporal criteria and those sensors.

The more stringent latitudinal criterion has proven to be a good choice for all comparisons, since latitudinal variations are, in general, more pronounced than longitudinal ones (Takele et.al., 2013). These criteria yielded 29, 77 and 118 days of coincident measurements between FTIR and MIPAS, MLS and AIRS respectively. The satellite data (MIPAS, MLS and, AIRS) used in the following comparisons have a considerably better vertical resolution than ground-based FTIR profiles due to observation geometry, spectral windows and measurement techniques.

-To a lesser extent, I also feel the authors have missed an opportunity to explain the aims of the study better, and the importance of the Addis Ababa site location and measurements made.

*The authors state that the comparisons at the "Addis Ababa station is good to study tropical atmospheric processes" (Pg 19, L12). „Good" in what context? Given the comparison results, will the ground-based CH4 and N2O measurements capture seasonal cycles and multi-year trends? Will biomass burning or other episodic events most likely be seen, and from what part of the tropics (the tropic is a large place)?*

Response on S2: Since this result is the first to the Addis Ababa site, it required to assess the quality of the ground-based FTIR measurements through comparing with data from MIPAS, MLS and AIRS sensors on board satellites. The following information is added at the end of the conclusion at pg 9, L 11.

The study has retrieved column abundances and profiles of two important green-house gases namely CH4 and N2O from solar absorption measurements taken at Addis Ababa, Ethiopia during a period that encompass May 2009 to March 2013. The fidelity of the data is assessed through comparison with data from MIPAS, MLS and AIRS satellites as well as full retrieval errors and their sources characterization. It is anticipated that the use of the data in further scientific studies may provide some insight into processes that govern chemical transport and chemistry in the atmosphere as well as sources of green gases in this part of the globe.

I am still willing to review/comment on any future manuscripts. It is a shame as the content and overall aim and methodology of the manuscript is robust and would be a welcome addition to the literature.

New references have added to the munscript:

[revised manuscript text omitted]

---

## Author Response (AR3)

Responses to referee 1:

We thank the referee for giving me another chance to see the manuscript.

Our responses are given point-by-point below (blue Times New Roman font) following each of the 'reviewers' comments, which are repeated in full (black Times New Roman Italic font). Reproduced text of the revised manuscript is set in green Times New Roman font.

In addition to the change made in the manuscript to take into account your comments or the comments of the other referees, several other changes might make and are shown the three documents (amt-2019-170-manuscript-version5.pdf, amt-2019-170-author_response-version2.pdf (with track changes) and amt-2019-170-supplement-version1.pdf). The page number and lines indicated on the lists are for commenting manuscript (amt-2019-170-manuscript-version5.pdf).

**Major comments:**

(1) *Authors attempt to reply major comment 1. However, in my opinion, it is not complete and/or thorough. The supplement shows a fit example on a single spectrum using micro-windows suggested by the NDACC/IRWG and the micro-windows adapted in this work. It only shows the measured and simulated spectra and residuals; however authors do not mention or compare columns, degrees of freedom, rms, sensitivity, effect of interference species, errors, etc. Showing only the residuals is not a justification that these five windows are suitable or better than the suggested by the IRWG/NDACC. In fact, Figure 1 (supplement) shows that CH4 absorption lines might now be well-isolated (panels 2 and 4) but it is hard to assess since no interference species are plotted. Are the CH4 absorption lines in the micro-windows used in this work well-isolated from other gases? most importantly, from H2O and HDO lines?. HDO might be an important interference but it is not even mention (I asked this before). Furthermore, I recommended to perform the analysis for several months and show an analysis, e.g., correlation of columns/rms, etc between IRWG/NDACC and retrieval setting applied in this work but authors do not mention or show this.*

Response: Details of the difference between the Micro-windows employed in the manuscript and the micro-windows recommended by NDACC are discussed for the Addis Ababa site in the supplementary document.

Here, the residuals, DOFs, the profile, sensitivity and error budget during the retrieval of CH4 from FTIR at Addis Ababa site have been used to justify retrieval setting applied in the manuscript.

HDO lines are interfering in the case where we use the microwindows recommended by RWG/DACC. Whereas, the HDO lines are not important in the microwindows applied in the manuscript and that is why only taking H2O lines.

For the last suggestions, I have analyzed them in the supplementary document (See Figure 6 and Table 2.)

(2) *Another major comment I had was about the comparison of profiles authors make between 11 to 27km, even though, for example for CH4, there is one degree of freedom in the troposphere and another in the stratosphere. The abstract and the manuscript sounds like authors claim more degrees of freedom. Authors response: "We took only the intercomparisons where a negative and positive bias are found to show the altitude ranges where volume mixing ratio derived from FTIR is higher (positive bias) and lower (negative bias). In the cases of DOFs, we have already stated in Pg 12, table 1.". It seems like authors have chosen signs in the bias to report range in altitude, however this might not be the best strategy. It would have been easier to compare upper troposphere lower stratosphere partial columns.*
Response: I think the problem might be happen at the time i wrote the response.

No, the signs of the bias don't have any relation with the altitude ranges selected for comparison. Whereas, the lower altitude determined from the sensitivity of MIPAS and the upper altitude from FTIR sensitivity ( See section 5.4). During the comparison of FTIR and MIPAS partial columns, we determined the DOFs over the altitude ranges where 15-27 km by taking the trace of averaging kernel with that altitude range. Hence, we didn't claim that more DOFs, i.e, 0 and 1.2 for CH4 and N2O respectively. (See page 17, Line 18)

In the cases of the altitude range employed during the intercomparison of the profiles; for comparison with AIRS, it goes between 11-27 km while the comparison with other instruments are 15-27 km as the sensitivity of the instruments are different.

(3) *From my last comments I asked why HITRAN 16 is not used and the response is:" This manuscript has been prepared before 3 years ago and even HITRAN 2004 was also not used in this work and rewritten as follows". It would have been very useful to know how HITRAN 16 compares with previous versions, especially for CH4. It happened to me so many times that I prepare a manuscript and then times fly and by the time I want to submit the abstract I actually have to update so many things, including time series, spectroscopy, etc. Authors mentioned that the manuscript was prepared three years ago, however this does not justify.*

Response: It would be difficult to do with HITRAN 16 for this manuscript, since my defense of the dissertation is near and i am also busy on it. Whereas, i promise to do it and sent you the result on another manuscript.

Responses to referee 2:

We thank the referee for giving me another chance to see the manuscript.

Our responses are given point-by-point below (blue Times New Roman font) following each of the 'reviewers' comments, which are repeated in full (black Times New Roman Italic font). Reproduced text of the revised manuscript is set in green Times New Roman font.

In addition to the change made in the manuscript to take into account your comments or the comments of the other referees, several other changes might make and are shown the three documents (amt-2019-170-manuscript-version5.pdf, amt-2019-170-author_response-version2.pdf (with track changes) and amt-2019-170-supplement-version1.pdf). The page number and lines indicated on the lists are for commenting manuscript (amt-2019-170-manuscript-version5.pdf).

**Comments on Authors reply/changes to the manuscript: amt-2019-170 for the Editor.**

*Yirdaw berhe, et al., have resubmitted their manuscript comparing ground-based MIR-FTS measurements of atmospheric CH4 and N2O at Addis Ababa, Ethiopia to that of three satellite (MIPAS, MLS and AIRS) data products.*

*I have received and read the authors reply/rebuttals (amt-2019-170-author_response-version2.pdf), along with the new supplement, amt-2019-170-supplement-version1.pdf. The authors supplied an updated/revised manuscript: amt-2019-170-manuscript-version5.pdf. The authors have made substantial changes to the manuscript to address reviewer's concerns.*

*Unfortunately, the revised manuscript presented at the end of the rebuttal document amt-2019-170-author_response-version2.pdf (which has highlighted changes, thanks, very handy) and the revised manuscript amt-2019-170-manuscript-version5.pdf (with no changes highlighted) are different. For example, section 2.2 is different between these two manuscript versions. Which is the correct version? This makes reviewing difficult. All commentary below is based upon the document: amt-2019-170-manuscript-version5.pdf*

Response: I appreciate the referee for checking both amt-2019-170-author_response-version2.pdf (with track changes) and amt-2019-170-manuscript-version5.pdf manuscript. Following the comment, I have corrected the statements in P 4, L 4-7 of "amt-2019-170-manuscript-version5.pdf" and similarly on the highlighted pdf document too.

The measured spectra have been analyzed using an algorithm that simulates the spectra and Jacobians by the line-by-line radiative transfer model PRFFWD (PRoFit ForWarD model) to produce the synthesized spectra and then, vertical profiles of CH4 and N2O would be derived by applying a retrieval code PROFFIT (Ver95) (Hase et al., 2004).

*Unfortunately (and not taken lightly), again, I cannot recommend the current altered manuscript (amt-2019-170-manuscript-version5.pdf) to be published. There are still too many spelling, grammatical and reference citation. Yirdaw berhe, et al. have addressed many of the initial reviewer's comments but I still find some critical replies/rebuttals are inadequate. Such inadequacies refer mainly to technical details, and once these are addressed (along with the mistakes) I would recommend publication. I do not see a major revision is required.*

Response: Thank you for your constructive comments you stated below and I will response to your comment point by point.

**Spelling, grammatical and reference citation mistakes:**

*Throughout the revised manuscript amt-2019-170-manuscript-version5.pdf and the reply to authors document (amt-2019-170-author_response-version2.pdf) there are numerous grammatical and spelling mistakes. The formatting of cited references still uses more than one type of style. I recommend the authors perform more intensive copy editing themselves or employ a copy-editing service.*

Response: I have corrected the reference style by applying \citep{}, \citet{}, \cite{}.and \citealt{}.Furthermore, the repetition of words in a consecutive sentence has been corrected.

**Technical concerns:**

1/ *The figures in amt-2019-170-supplement-version1.pdf are of a low visual standard. Could the figures be replaced with those of a correct aspect-ratio?*

Response: I have added different information on the supplement document so that to show clearly the differences of the micro-windows used in the manuscript from the micro-windows recommended by NDACC (As suggested by the second referee). Furthermore, the previous figure has also been corrected (please see the supplement document).

2/ Retrieval strategy and quality:

*The theory of the Tikhonov retrieval strategy is provided in the supplement and (starting) on pg. 5, L22 of the revised manuscript. The details supplied do not address the questions of the reviewers on this topic. It is stated at pg. 6, ln 1 in the revised manuscript "An optimised retrieval strategfy for tropics has been established", but there are no details given on the method or results of the optimised strategy. For example, what is the value of the Alpha parameter and how was this value decided upon. What is the Se value used? Also, the retrieval strategy is also optimised for Addis Ababa, not the for tropics in general.*

Response: The optimized strategy employed in the manuscript starts from selecting all settings (micro-windows, constraint, initial guess and a priori profile) to the Addis Ababa site in such a way that all the structures visible in the retrieved distributions originate from the measurements. (See details in the supplementary document). The value for $S_e$ is taking the identity I and the site has also expressed as Addis Ababa.

Response on Pg 5/L10: The information below has been inserted in pg 5/L10 of the first manuscript: (se also the supplementary document on retrieval set up).

PROFFIT includes various retrieval options such as scaling of a priori profile, the Tikhonov-Phillips method (Phillips, 1962; Tikhonov, 1963), or the optimal estimation method (Rodgers , 2000). In this study, an optimized retrieval strategy for Addis Ababa has been established for the retrieval of CH4 and N2O by applying it first to single spectra, as test cases, and later routinely to the full set of measurements. Part of the strategy to optimally retrieval of the total columns of CH4 and N2O are to search for a set of spectral micro-windows, constraint, initial guess and a priori profile are chosen in such a way that all the structures visible in the retrieved distributions originate from the measurements and are not artifacts due to any constraints. At the Addis Ababa site, we did not use the a priori covariance matrix as an optimal estimation. However, the Tikhonov-type $L_1$ regularization method (Sussmann et.al., 2009) on a logarithmic scale is used during the retrieval of CH4 and N2O. The retrieval is performed on a fine vertical grid from 2.45 to 85 km and is stabilized by a first order Tikhonov constraint, R= $\alpha L^T_1 L_1$, where $\alpha$ is the strength of the constraint and $L_1$ is the first order derivative (Borsdorff et al., 2014), which smooths the solution without biasing it towards the a priori profile. The parameter determines the weight of the regularization and it is also important to choose appropriate to the problem. One way to fix this parameter is the L-curve method (Hansen, 1992). The regularization strength $\alpha$, is determined by finding a trade-off between the number of degrees of freedom (a measure of the amount of information in methane and nitrous oxide retrieval), which is given by the trace of the averaging kernel and the noise induced error (Rodgers, 2000). A regularization strength $\alpha$, of 2.5 $X10^4$ was found optimum for CH4 retrieval.

*On pg. 6, ln 7 the manuscript states "The magnitude of the residuals indicates that measured spectra….of both CH4and N2O was quality as they are less than 1". Apart from grammatical mistake, the authors apply a QC/QA filtering limit of '1' to the residuals to assess retrieval 'quality'. There are no units of this value, or a description of the meaning of the value. Could the authors please explain in more detail the methods assessing retrieval quality.*

Response: Thank you for your suggestion, the statement has changed as follows.

The magnitude of residuals indicates systematic errors in the spectroscopic line data used to derive the concentration of CH4 and N2O. Therefore, the fits are good with an averaged root mean square residual of 0.12 % for the micro windows selected in the retrieval of CH4.

The following information has added on pg-6 L7 to make clear how the quality would assess.

The quality of the measurements during the time period of May 2009-February 2011 for ozone has been revealed by Takele Kenea et al. (2013). Whereas, the quality of the measurements of CH4 and N2O has also assessed through the sensitivity, DOFs and the contribution of different error sources on measurements in addition to the spectral residuals that indicate systematic errors in the spectroscopic line data.

*On pg. 4, ln 25 the statement "The quality of the measurements during the time period of May 2009-February 2011 has revealed by Takele Kenea et al. (2013)." is out of place in this section and has no context. This statement should be at the end of section 3.1 and the authors should also give the readers more insight as to what the actual criteria are used to assess quality.*

Response: We have corrected it as stated in the above response.

3/ *I think the authors response and manuscript correction to the reviewer's question below is inadequate. My understanding is that measurements started in 2009 to present, is this correct? Why you use limited number of years? I highly encourage to include more years and if possible trend analysis. Do the trend make sense?*

Response on P10, L8: It is difficult to show the trend analysis since the data has more discontinuity during the measurements. The data analyzed in this paper covers only for period time of May 2009 to March 2013 with a number of months and days missing during the period due to technique problem. The instrument has not operated after March 2013 for a long period and we prepared the manuscript for those data sets only. However, The instrument is operated at 2016/2017 and ceased after 2017.

*The authors need to explicitly state why certain periods were excluded, i.e. the instrument was not functional between March 2011 and November 2012. Also, why comparisons halted at March 2013. Was the instrument not working from 2013 onwards? Or alternatively, why the period May 2009 to March 2013 was the period selected for satellite comparisons.*

Response: The number of days selected for the satellites depends on the sensitivity of the instruments, for instance in the case of MIPAS, the lower altitude range obtained between 15-20 km and above and we ignored the data with lower altitude sensitive not below 18 km. MIPAS data provider has released only for time period up to March 2012.

The manuscript has prepared for this period, as the instrument has not operated for a long time after March 2013. However, the instrument had also operated in 2015 and 2016.

4/ Coincidence criteria.

*The authors state "The more stringent latitudinal criterion has proven to be a good choice for all comparisons, since latitudinal variations are, in general, more pronounced than longitudinal ones" (pg. 12, ln 9). Could the authors explain why the chosen limits are more stringent (and compared to what?) and why they are a good choice.*

Response: As i have mentioned in the manuscript, the spatial criteria that is the latitudinal and longitudinal were varied ±20 for latitude and ±100 for longitude. The reason why the spatial criteria narrow range for latitude and long rang for longitude has been considered in this work is due to the variability of the parameters along latitude is high as compared to the

longitudinal variation. (I try to show the reason why this spatial criterion has used in Figure 6 of the supplementary document). In the previous works of the site, also use such criteria.

5/*WACCM citable reference: Pg. 4, ln 31. The WACCM model needs a citable reference. http://www2.cesm.ucar.edu/working_groups/?ref=navdoes not refer to WACCM.*

Response: I corrected the sentences and adding the citable reference.

The a prior $x_a$ for methane and the interfering species above Adiss Ababa were taken from 40 yr averages (1980–2020) of the Whole Atmosphere Climate Chemistry Model (WACCM, Garcia et al., 2007). Similarly, the a priori nitrous oxide profile has also constructed from monthly averages data available from WACCM (e.g., Tilmes et al., 2007). Whereas, the grid to be used for Addis Ababa site is found with the WACCM mixing ratio profile data at "ftp://ftp.acom.ucar.edu/user/jamesw/IRWG/2013/WACCM/V6/Addis_Ababa/"

6/ *pg.12 ln 18: "Hence, the profiles from MIPAS, MLS and AIRS have been degraded to make a comparison between the FTIR and satellite observations." I think degraded is the wrong word, ambiguous. Could the authors rewrite or explain the term 'degraded' in a better way.*

Response: The word "degraded" has substituted by "smoothed"

7/ *Partial column altitude range in section 5.4 and Fig 11. Caption I infer the partial column range is 15-27km? is this correct? Could the authors please explicitly state the exact range in section 5.4 and fig 11 caption.*

Response: Yes, we added information about altitude range in section 5.4 and caption of figure 11.

"for altitude range 15-27 km"

[revised manuscript text omitted]